# EMR-AGENT: Automating Cohort and Feature Extraction from EMR Databases

## Abstract

Machine learning models for clinical prediction rely on structured data extracted from Electronic Medical Records (EMRs), yet this process remains dominated by hardcoded, database-specific pipelines for cohort definition, feature selection, and code mapping. These manual efforts limit scalability, reproducibility, and cross-institutional generalization. To address this, we introduce EMR-AGENT (Automated Generalized Extraction and Navigation Tool), an agent-based framework that replaces manual rule writing with dynamic, language model-driven interaction to extract and standardize structured clinical data. Our framework automates cohort selection, feature extraction, and code mapping through interactive querying of databases. Our modular agents iteratively observe query results and reason over schema and documentation, using SQL not just for data retrieval but also as a tool for database observation and decision making. This eliminates the need for hand-crafted, schema-specific logic. To enable rigorous evaluation, we develop a benchmarking codebase for three EMR databases (MIMIC-III, eICU, SICdb), including both seen and unseen schema settings. Our results demonstrate strong performance and generalization across these databases, highlighting the feasibility of automating a process previously thought to require expert-driven design. The code will be released publicly. For a demonstration, please visit our anonymous demo page: https://anonymoususer-max600.github.io/EMR_AGENT/

## 1 Introduction

Electronic Medical Records (EMRs) encapsulate diverse patient-related data, including patient insurance, demographics, vital signs, lab results, clinical images, and clinical notes. Recent advances in machine learning (ML) have accelerated the development of predictive models using these various EMR data (Horn et al., 2020; Li et al., 2023; Luo et al., 2024; Shukla & Marlin, 2021; Tipirneni & Reddy, 2022). Leveraging this rich clinical information, ML models are increasingly employed to support timely interventions and optimize resource allocation, with the goal of preventing patient clinical deterioration and improving patient outcomes (Lee et al., 2023a;b). However, ensuring reproducibility and comparability of these models necessitates consistent preprocessing steps, particularly for cohort selection, feature selection (*e.g.*, age, gender, mortality status), and code mapping of clinical measurements (*e.g.*, laboratory test results, vital signs).

In practice, these preprocessing steps are manually crafted and closely tied to each hospital's EMR schema, hindering scalability and reuse across different institutions (Hur et al., 2022; Jarrett et al., 2021; McDermott et al., 2021). Specifically, two significant challenges arise from EMR-side factors:

First, semantic and structural heterogeneity is common across EMR systems from different manufacturers and institutions. Hospitals significantly differ in how they structure, store, and annotate clinical data. For example, the variable "heart rate" may appear as "itemid=211" in MIMIC-III (a large single-center ICU database from the U.S. (Johnson et al., 2016)), "HeartRateECG" in SICdb (a European ICU dataset (Rodemund et al., 2023)), or as a column "heartrate" in eICU (a multi-center ICU dataset from the U.S). Extending this complexity to real-world clinical settings further complicates the picture, as actual hospital EMRs often contain different schemas designed independently by various EMR system manufacturers (Gamal et al., 2021; Hamadi et al., 2022; Wornow et al., 2023). Consequently, ML models trained on data from one EMR system often exhibit poor comparability and generalizability when deployed on datasets from different EMR systems, as

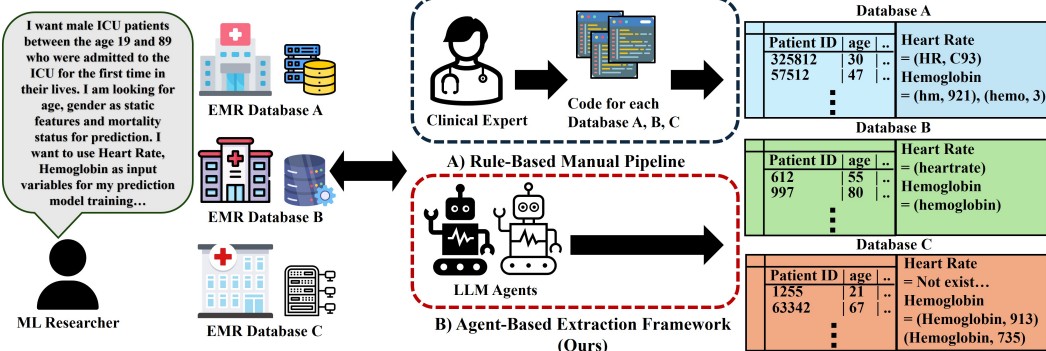

**Figure 1:** Illustration of the shift from (A) the conventional **Rule-Based Manual Pipeline**, where clinical experts must handcraft cohort and feature extraction logic as well as mapping codes for each database, to (B) our **EMR-AGENT (Agent-Based Extraction Framework)**, which automates these processes through iterative interaction with the database, enabling generalization to diverse schemas.

variations in schema structures and data annotations significantly impact model input consistency (Hur et al., 2022). Several harmonization frameworks-such as YAIB (van de Water et al., 2024), ACES (Xu et al., 2025), Clairvoyance (Jarrett et al., 2021), ES-GPT (McDermott et al., 2023), and BlendedICU (Oliver et al., 2023)—have attempted to address these heterogeneities. However, these frameworks remain either too rigid due to hard-coded, dataset-specific rules (YAIB, BlendedICU) or overly dependent on predefined input formats (ACES, Clairvoyance, ES-GPT), limiting their flexibility and generalizability.

Second, variability persists even within the same EMR dataset, due to inconsistent code mappings and cohort selection procedures. Clinical concepts such as heart rate can be measured through multiple modalities (*e.g.*, sensor data, auscultation, palpation), resulting in numerous potential code mappings (Oliver et al., 2023). Additionally, cohort selection processes are often subjective, as selection instructions can be interpreted differently across studies or research groups. For instance, an instruction such as "include patients admitted to the ICU for the first time" might ambiguously include or exclude patients with previous ICU stays, depending on researcher interpretation (Harutyunyan et al., 2019; Purushotham et al., 2018; Wang et al., 2020; Wornow et al., 2023). Even when criteria are clear, clinical experts have to hard code them separately for each database due to their heterogeneous nature. These ambiguities and inconsistencies force researchers to reverse-engineer database schemas and craft bespoke preprocessing pipelines for each study.

In this work, we introduce the first AI-based EMR preprocessing framework, named **EMR-AGENT (Automated Generalized Extraction and Navigation Tool)**, that automates structured data extraction - including cohort selection, feature identification, and code mapping - without manual rule crafting or expert intervention. As illustrated in Fig. 1, EMR-AGENT leverages large language model (LLM) agents that actively interact with live EMR databases, observe query outputs, and reason over schema and documentation to guide the extraction process. Unlike conventional Text-to-SQL approaches (Jo et al., 2024; Marshan et al., 2024; Pourreza & Rafiei, 2023; Ryu et al., 2024; Talaei et al., 2024), our agents treat SQL not merely as an endpoint but as a means for iterative exploration, validation, and decision-making.

Our contributions are summarized as follows:

- We propose EMR-AGENT, the first LLM-based framework, composed of the Cohort and Feature Selection Agent (CFSA) and the Code Mapping Agent (CMA), for essential EMR preprocessing tasks without manual rules or expert input.
- To rigorously evaluate automated EMR preprocessing capabilities of our framework, we construct dedicated benchmark suites for three ICU databases-MIMIC-III, eICU, and SICdb. These benchmarks assess the agent's ability to extract relevant patient cohorts from user-defined clinical requests and standardize mapping codes across different database schemas.
- We demonstrate the generalization and robustness of EMR-AGENT through extensive experiments, including (1) component-level ablation studies, (2) comparison against alternative LLM-based approaches, and (3) evaluations on previously unseen EMR databases, showing that our framework can achieve results comparable to human experts in new cohort and feature selection tasks.

## 2 RELATED WORK

### 2.1 BENCHMARK FRAMEWORKS FOR EMR PREPROCESSING

Numerous clinical prediction models have been developed using EMR data for tasks such as in-hospital mortality, decompensation, and length of stay (Horn et al., 2020; Li et al., 2023; Luo et al., 2024; Shukla & Marlin, 2021). These models typically rely on dataset-specific preprocessing pipelines with custom inclusion criteria and variable extraction logic (*e.g.*, MIMIC-Extract (Wang et al., 2020), Harutyunyan et al. (Harutyunyan et al., 2019), eICU-Benchmark (Sheikhalishahi et al., 2020), the PhysioNet Challenge (Goldberger et al., 2000), Reyna et al. (2019), and EHRSHOT (Wornow et al., 2023)). As each benchmark encodes different assumptions about cohort selection and variable composition, even models trained on the same base dataset (*e.g.*, MIMIC-III) yield divergent patient populations and extracted features (Harutyunyan et al., 2019; Purushotham et al., 2018; Wang et al., 2020), complicating fair comparison and reproducibility (McDermott et al., 2021). This fragmentation also hinders the development of general-purpose foundation models for EMRs, as well as making it difficult to establish cross-domain evaluation or domain generalization method on EMRs, demanding additional efforts by human experts.

To address this, several harmonization frameworks aim to enable multi-database compatibility. BlendedICU (Oliver et al., 2023) and YAIB (van de Water et al., 2024) provide expert-curated cohort definitions and mappings but are tightly coupled to specific datasets through handcrafted rules, limiting generalizability. ACES (Xu et al., 2025) introduces a flexible task configuration language but still requires specific data formats (*e.g.*, MEDS (Arnrich et al., 2024), ES-GPT (McDermott et al., 2023)), necessitating additional preprocessing. Clairvoyance (Jarrett et al., 2021) and ES-GPT provide modular pipelines but likewise depend on fixed input formats, with MEDS offering a standardized event-based schema that still requires dataset conversion. While these tools improve intra-dataset consistency, adapting them to new institutions or clinical features remains challenging due to their reliance on fixed formats or handcrafted rules.

### 2.2 AI INTERACTION WITH EMR DATABASES

Recent LLM-based Text-to-SQL models for EMR databases, such as PLUQ (Jo et al., 2024), EHR-SeqSQL (Ryu et al., 2024), and MedT5SQL (Marshan et al., 2024), primarily translate clinical questions into SQL queries. These models assume that users—typically doctors or clinicians—are familiar with the database schema, implying a direct correspondence between the query and the schema (*e.g.*, the word "drug" in EHRSQL 2024 (Lee et al., 2022) directly maps to the column name `drug` in their EMR database). However, these architectures lack dynamic database interaction capabilities and cannot handle schema ambiguities, limiting their applicability for complex EMR preprocessing tasks. Moreover, real-world EMR databases often exhibit complex and variable schemas across hospitals, making the assumption of prior schema knowledge unrealistic. Consequently, the lack of dynamic interaction and schema variability hinders the robustness of current EMR preprocessing systems.

Agent-based frameworks like Spider 2.0 (Lei et al., 2025) introduce SQL-query based interactive, multi-turn reasoning with databases, including error correction. EHRAgent (Shi et al., 2024) extends this idea to EMR settings by executing SQL over real EHR data. However, both approaches focus on answering isolated queries (*e.g.*, chart review) rather than automating structured preprocessing. In contrast, EMR preprocessing-such as cohort selection or code mapping-requires iterative observation, reasoning across heterogeneous schemas, and verification via query outputs. In these settings, SQL is a means of exploration, not a final output. As such, existing text-to-SQL systems are insufficient for building generalizable EMR preprocessing pipelines.

## 3 PROPOSED FRAMEWORK: *EMR-AGENT*

In this section, we introduce our framework, **EMR-AGENT**, the first AI-driven solution for automated preprocessing of electronic medical records (EMRs) covering cohort selection, feature extraction, and code mapping as illustrated in Fig. 2.

Traditional preprocessing pipelines for EMR databases - *e.g.*, vital signs, and lab test results - have largely remained reliant on rule-based methods, typically requiring manual curation by domain

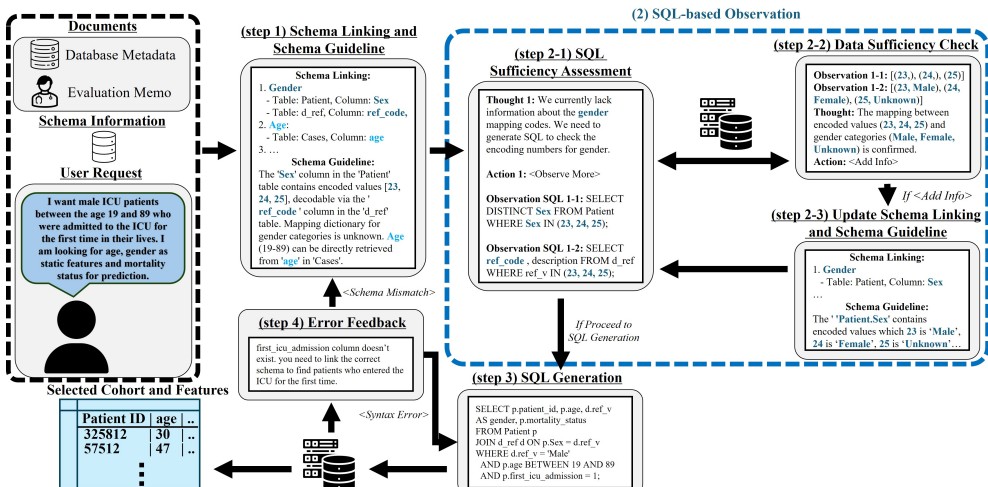

**(a)** Cohort and Feature Selection Agent: Automates the process of selecting cohorts and relevant features from heterogeneous databases through an iterative interaction framework.

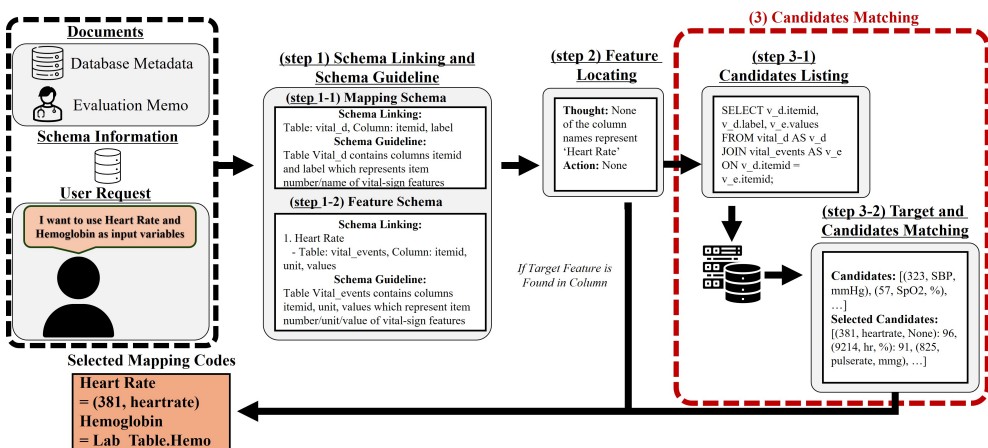

**(b)** Code Mapping Agent: Standardizes feature representation by mapping database-specific codes.

**Figure 2:** Illustration of the two main components of EMR-AGENT: (a) CFSA dynamically selects cohorts and features from diverse EMR databases, reducing manual intervention; (b) CMA harmonizes database-specific codes for uniform feature representation.

experts (Goldberger et al., 2000; Harutyunyan et al., 2019; Sheikhalishahi et al., 2020; van de Water et al., 2024; Wang et al., 2020; Xu et al., 2025). EMR-AGENT overcomes this bottleneck with two LLM-based agents: (1) the **Cohort and Feature Selection Agent (CFSA)** (Section 3.2), responsible for extracting patient cohorts and clinical variables, including demographics and clinical events, and (2) the **Code Mapping Agent (CMA)** (Section 3.3), designed to standardize clinical feature codes for vital signs and lab tests across heterogeneous EMR systems. Both agents adopt a problem decomposition strategy, breaking complex tasks into manageable sub-problems (Pourreza & Rafiei, 2023; Shi et al., 2024; Wei et al., 2022).

Each agent starts with the **Schema Linking and Guideline Generation** step (Section 3.1). To fulfill the user request, relevant schema metadata, database manuals, and evaluation notes are selectively retrieved. Based on this schema-linked information, a guideline is generated that explains the linked schema, plans how to execute the user request via SQL, and identifies what is missing information required for the execution. Armed with this guideline, both CFSA and CMA dynamically execute SQL queries on the EMR database to gather missing or necessary information and complete the preprocessing stage, as described in the following subsections. This structured process mirrors clinical practice, where professionals first familiarize themselves with the EMR documentation to

identify the desired cohort, features, or codes, and then explore the EMR database to complete the task with a deeper understanding.

**Inputs of agents**   Both agents process three types of input: *user-defined clinical requests*, *documents*, and *schema information*. Clinical requests, written in natural language, specify the desired patient cohort, features, or clinical variables. The documents include the EMR database manual, a human-curated guide detailing the database's structure and semantics, and evaluation memos, concise notes from clinical experts highlighting dataset caveats. Schema information comprises the list of tables and columns, along with $N$ sample values per column, providing concrete insight into the data structure.

## 3.1   Schema Linking and Guideline Generation

The CFSA and CMA begin with the **Schema Linking and Guideline Generation** module (Fig. 2a, Fig. 2b). Unlike traditional schema linking (Lei et al., 2020; Pourreza & Rafiei, 2023), which relies solely on schema information, our approach leverages multiple knowledge sources given as documents, including database manuals and evaluation memos, to enhance schema linking.

To effectively manage information from diverse sources, we introduce the **Schema Guideline** method. This method systematically specifies the role and usage of each linked table and column while identifying any missing or ambiguous elements that require further verification to fulfill the user request. Unlike planning-based web automation methods (Gu et al., 2024; Gur et al., 2024), which primarily decompose tasks into smaller steps, the Schema Guideline method focuses on identifying information gaps.

In CFSA, the Schema Guideline clarifies each schema component's role and highlights missing information, allowing the agent to plan SQL-based observation (Section 3.2). This makes schema linking context-aware and practical for subsequent SQL generation, even when column names and sample values lack clarity. For instance, as shown in step 1 from Fig. 2a, the Schema Guideline identifies the absence of gender code information, indicating that SQL generation is not yet feasible.

In contrast, CMA uses the Schema Guideline to define the role of each table and column for accurate candidate listing (Section 3.3). It ensures that only verified schema information is used for precise SQL generation for candidates listing. For example, in step 1 from Fig. 2b, the Schema Guideline identifies columns representing the item number of vital signs, essential for candidates listing.

## 3.2   Cohort and Feature Selection Agent (CFSA)

The CFSA comprises three core components beyond schema linking: **SQL-based Observation**, **SQL Generation**, and **Error Feedback** (Fig. 2a, Appendix B.1).

**SQL-based Observation**   ensures the sufficiency of the linked schema and guideline by interacting with the EMR database as needed. It consists of three steps:

- **SQL Sufficiency Assessment:** Determines whether the current schema and guideline can generate the desired SQL. If inadequate, the agent formulates observation SQL queries to gather additional data (*e.g.*, sample values) from the live EMR database. If sufficient, it proceeds to SQL generation. For instance, in 2-1 step from Fig. 2a, CFSA searches for male patients but, lacking gender data format, generates multiple SQL queries to identify how it is stored in the target EMR database.
- **Data Sufficiency Check:** After executing the observation SQL queries, the agent evaluates whether the retrieved data improves the schema and guideline. If informative, it moves to the **Schema Update**; otherwise, it repeats the sufficiency assessment. For example, in 2-2 step from Fig. 2a, CFSA discovers that the reference code for "Male" is 23, a critical piece of information.
- **Schema Update:** Integrates newly obtained data into the schema linking and guideline, addressing any previously incomplete information in both linked schema and guideline.

In the **SQL Generation** step, CFSA generates queries using the refined schema and guidelines. The **Error Feedback** module classifies the SQL outputs into three categories:

- **Syntactic Error:** SQL queries with syntax errors are immediately regenerated.
- **Schema Mismatch:** SQL queries that are syntactically valid but produce semantic errors (*e.g.*, empty outputs, missing columns, invalid types). In such cases, the agent returns to the **Schema Linking and Guideline Generation** step, incorporating the error message as feedback.

- **Correct Result:** When the query executes successfully and returns valid outputs, the agent finalizes the extraction of the requested cohort and features.

This feedback loop is retried up to the maximum number of attempts specified in Section 5.1, enabling the agent to recover from both explicit and subtle schema inconsistencies without manual debugging.

## 3.3 Code Mapping Agent (CMA)

Similar to CFSA, CMA begins with **Schema Linking and Guideline Generation**, with its primary goal being to map user-requested variables to mapping codes from the EMR database. It includes two core modules: **Feature Locating** and **Candidates Matching** (Fig. 2b, Appendix B.2).

**Feature Locating** initially searches for the requested feature directly as a column name from linked schema. If the feature is found, it returns the corresponding table and column names. If not, the agent assumes that the feature may either be stored as a row value or may not exist in the current EMR database, and proceeds to **Candidates Matching**.

**Candidates Matching** The process begins with **Candidates Listing**, where the agent identifies potential tables and columns from the linked schema that may contain the ID, feature name, and unit of the user-requested feature. It then generates SQL `DISTINCT` queries to retrieve all candidate combinations from the identified columns. After preparing the candidate list, the agent proceeds to the **Target and Candidates Matching** step, where it compares the user-requested feature with candidates in batches, calculating similarity scores and retaining only those that exceed the predefined threshold. For example, in step 3-2 of Fig. 2b, the candidates listed are compared with the user-requested feature, and CMA evaluates the similarity score (0 to 100). The final candidates are determined based on the similarity threshold, a hyperparameter set by the user. By adjusting this threshold, the user can lower it to increase recall, even at the cost of precision. This user-controlled threshold adds practicality, allowing users to balance recall and precision according to their needs. Lowering the threshold increases recall by capturing more candidates, while raising it reduces false positives, prioritizing precision.

## 4 EMR Preprocessing Benchmark: *PreCISE-EMR*

In addition to proposing the LLM-driven EMR preprocessing framework, we also aim to address the notable lack of standardized evaluation protocols for such tasks. As existing benchmarks primarily focus on downstream task performance rather than the data acquisition side, we construct a standardized evaluation protocol and codebase tailored for rigorous assessment of EMR preprocessing quality, named *PreCISE-EMR* (**Pre**processing for **C**ohort **I**dentification, Feature **S**election, and Code **E**xtraction, in collaboration with clinical experts.

### 4.1 Database Environment Setup

We use three publicly available EMR datasets: MIMIC-III (v1.4) (Johnson et al., 2016), eICU (v2.0) (Pollard et al., 2019), and SICdb (v1.0.8) (Rodemund et al., 2023) (Table A.1). These datasets are set up with the official open-source scripts[1][2], which ensure consistent data processing and loading into PostgreSQL environments while preserving the original schema. For SICdb, we manually convert the provided CSV files into PostgreSQL. The resulting environments are used to generate evaluation sets comparing EMR-AGENT's outputs with human judgments.

Notably, since the release dates of MIMIC-III in September 2016, eICU in April 2019, and SICdb in September 2024, we consider SICdb as an *unseen* EMR database in our experiment. This distinction is based on the knowledge cutoff date (June 2024) of the primary backbone LLM we used (Claude-3.5-Sonnet (Anthropic, 2024a)), indicating that SICdb was not part of its training data. Additionally, compared to MIMIC-III (26 tables) and SICdb (7 tables), eICU's schema is more intricate with 31 tables and features appearing as both column names and row values, making schema parsing and data extraction more challenging.

---

[1]MIMIC-III: https://github.com/MIT-LCP/mimic-code/tree/main/mimic-iii/buildmimic/postgres

[2]eICU: https://github.com/MIT-LCP/eicu-code/tree/main/build-db/postgres

**Table 1:** Performance comparison of (EMR) Text-to-SQL methods and the Agent-based method to our approach. Results include the average F1 score and (balanced) accuracy with standard error for (a) Cohort and Feature Selection and (b) Code Mapping.

| (a) Cohort and Feature Selection | | | | | |
|---|---|---|---|---|---|
| **Method** | MIMIC-III | | eICU | | SICdb | |
| | F1 | Acc. | F1 | Acc. | F1 | Acc. |
| Ours | **0.94**$\pm$0.01 | **0.893**$\pm$0.01 | **0.929**$\pm$0.03 | **0.951**$\pm$0.03 | **0.814**$\pm$0.04 | **0.794**$\pm$0.04 |
| ICL(PLUQ) (Jo et al., 2024) | 0.749$\pm$0.04 | 0.809$\pm$0.04 | 0.132$\pm$0.04 | 0.131$\pm$0.04 | 0.407$\pm$0.02 | 0.428$\pm$0.02 |
| ICL(SeqSQL) (Ryu et al., 2024) | 0.04$\pm$0.01 | 0.173$\pm$0.04 | 0.00$\pm$0.0 | 0.00$\pm$0.0 | 0.04$\pm$0.04 | 0.08$\pm$0.05 |
| DinSQL (Pourreza & Rafiei, 2023) | 0.726$\pm$0.05 | 0.72$\pm$0.04 | 0.00$\pm$0.0 | 0.00$\pm$0.0 | 0.071$\pm$0.03 | 0.036$\pm$0.02 |
| REACT (Yao et al., 2023) | 0.308$\pm$0.05 | 0.308$\pm$0.04 | 0.524$\pm$0.06 | 0.542$\pm$0.06 | 0.503$\pm$0.04 | 0.493$\pm$0.03 |

| (b) Code Mapping | | | | | |
|---|---|---|---|---|---|
| **Method** | MIMIC-III | | eICU | | SICdb | |
| | F1 | bAcc. | F1 | bAcc. | F1 | bAcc. |
| Ours | **0.516**$\pm$0.0 | **0.283**$\pm$0.01 | **0.648**$\pm$0.05 | **0.336**$\pm$0.03 | **0.536**$\pm$0.03 | **0.38**$\pm$0.02 |
| ICL(PLUQ) (Jo et al., 2024) | 0.022$\pm$0.01 | 0.036$\pm$0.0 | 0.125$\pm$0.01 | 0.112$\pm$0.01 | 0.119$\pm$0.0 | 0.078$\pm$0.00 |
| REACT (Yao et al., 2023) | 0.214$\pm$0.05 | 0.14$\pm$0.01 | 0.067$\pm$0.0 | 0.081$\pm$0.0 | 0.218$\pm$0.0 | 0.154$\pm$0.00 |

## 4.2 Ground-truth Construction

**Cohort and feature selection** We define evaluation sets focusing on harmonizability and reliability. Harmonizability ensures that our agent consistently selects the same patient groups and features across three heterogeneous databases, enabling the creation of compatible datasets for downstream models. To achieve this, we construct a Cohort and Feature Selection evaluation set by varying exclusion criteria (*e.g.*, age, gender, minimum clinical records, etc.) to generate multiple complex cohorts (Table A.2) on the varying EMR databases. Reliability is assessed by verifying whether our benchmark code, when using the same cohort criteria, selects the same patient groups as existing benchmarks (Harutyunyan et al., 2019; Sheikhalishahi et al., 2020) (Fig. A.1, A.2).

**Code mapping** Following the approach of detailed evaluation memos (Fig. A.3), we select a total of 56 features, limited to vital signs and laboratory results (Table A.3). All features are searched in the Athena Observational Health Data Sciences and Informatics databases (ATHENA, 2023) for clinical concepts and are defined using standard terminology. For each dataset, a team consisting of two medical doctors, two nurses, and one clinical expert conduct feature mapping processes, create a mapping dictionary that serves as the ground truth for evaluating code mapping (Fig. A.4).

## 4.3 Evaluation Process

To assess the EMR preprocessing accuracy of EMR-AGENT, we use our newly constructed evaluation sets for cohort and feature selection task and mapping dictionary for code mapping task, respectively. The CSFA is evaluated by comparing ICU stay's ID from evaluation sets with agent's results, averaging the performance over 10 repeated trials. Generally, error cost priorities can vary across different clinical contexts. In our clinical research subject selection scenario, minimizing false negatives takes priority, as missing eligible patients causes a greater risk, while maintaining high precision among selected subjects is also crucial to ensure the accuracy of identified candidates. Based on these clinical objectives, we adopt the F1 score as our evaluation metric, as it effectively balances recall and precision. We additionally evaluate the accuracy of required format for demographic and clinical variables (gender, age, mortality status, and length of stay). For the CMA, we conducted evaluation by comparing the mapping dictionary with agent's results, averaging the performance over 3 repeated trials. We used both the F1 score and balanced accuracy as metrics to provide a balanced assessment of mapping quality. Note that PreCISE-EMR is a benchmarking framework, not a dataset. It requires users to obtain appropriate credentials (*e.g.*, via PhysioNet) and execute the code locally; thus, no derived patient-level data are redistributed.

## 5 Experiments

 In this section, we provide the detailed experimental setup and evaluation protocols used to assess the performance of our proposed EMR-AGENT. We present the performance evaluation of our

**Table 2:** Ablations of (a) CFSA - F1/Accuracy drop from component removal, (b) CMA - F1/Balanced Accuracy drop from disabling Candidate Matching and SchemaGuideline. DB Interact* represents both SQL-based Observation/Error Feedback.

| (a) Cohort and Feature Selection | | | | | | |
|---|---|---|---|---|---|---|
| **Method** | MIMIC-III | | eICU | | SICdb | |
| | F1 | Acc. | F1 | Acc. | F1 | Acc. |
| Ours | **0.94**$\pm$0.01 | **0.893**$\pm$0.01 | **0.929**$\pm$0.03 | **0.951**$\pm$0.03 | **0.814**$\pm$0.04 | **0.794**$\pm$0.04 |
| Ours w/o SQL-based Observation | 0.916$\pm$0.01 | 0.881$\pm$0.01 | 0.898$\pm$0.03 | 0.951$\pm$0.03 | 0.795$\pm$0.05 | 0.602$\pm$0.04 |
| Ours w/o Error Feedback | 0.688$\pm$0.05 | 0.668$\pm$0.05 | 0.624$\pm$0.06 | 0.642$\pm$0.06 | 0.617$\pm$0.06 | 0.572$\pm$0.05 |
| Ours w/o DB Interaction* | 0.677$\pm$0.05 | 0.648$\pm$0.05 | 0.562$\pm$0.06 | 0.57$\pm$0.06 | 0.57$\pm$0.06 | 0.428$\pm$0.05 |
| Ours w/o SchemaGuideline | 0.827$\pm$0.03 | 0.825$\pm$0.01 | 0.87$\pm$0.03 | 0.892$\pm$0.04 | 0.792$\pm$0.05 | 0.692$\pm$0.04 |
| (b) Code Mapping | | | | | | |
| **Method** | MIMIC-III | | eICU | | SICdb | |
| | F1 | bAcc. | F1 | bAcc. | F1 | bAcc. |
| Ours | **0.516**$\pm$0.0 | 0.283$\pm$0.01 | **0.648**$\pm$0.05 | **0.336**$\pm$0.03 | **0.536**$\pm$0.03 | **0.38**$\pm$0.02 |
| Ours w/o Candidates Matching | 0.0$\pm$0.0 | 0.0$\pm$0.0 | 0.07$\pm$0.0 | 0.035$\pm$0.0 | 0.0$\pm$0.0 | 0.0$\pm$0.0 |
| Ours w/o SchemaGuideline | 0.508$\pm$0.0 | **0.285**$\pm$0.02 | 0.575$\pm$0.02 | 0.329$\pm$0.0 | 0.342$\pm$0.01 | 0.209$\pm$0.01 |

proposed approach in four key areas: 1) comparison with baseline methods, 2) component ablation of CFSA and CMA, 3) external knowledge impact, and 4) performance variation across different LLM models. We use our own benchmark described in Section 4. Unless otherwise specified, we employ Claude-3.5-Sonnet (Anthropic, 2024a) as the backbone LLM. Importantly, its use fully complies with Data Use Agreement (DUA) of PhysioNet, and all experiments in this study were conducted in strict adherence to these requirements PhysioNet (2023).

## 5.1 EXPERIMENT SETUP

**Baselines** Since the task we address is novel and has not been previously considered, there are no direct baselines available. Although the objectives of existing models differ somehow from ours, we select the most relevant approaches to demonstrate that even their naive application cannot easily solve our task: PLUQ-prompt-style LLM for text-to-SQL tasks (Jo et al., 2024); multi-turn SeqSQL for sequential SQL generation based on EHR-SeqSQL (Ryu et al., 2024); DIN-SQL, which decomposes text-to-SQL into modular steps like schema linking and SQL type classification (Pourreza & Rafiei, 2023); and REACT, an agent-based method for dynamic query generation (Yao et al., 2023). All baselines are provided with schema information and external knowledge, including database metadata and evaluation memos. We adapt each baseline prompt to the PostgreSQL setting.

**Hyperparameter setting** Due to token limits, schema information includes 10 sample values per column. CFSA allows up to 10 observations (5 queries per observation), with temperature set to 0 for the first 5 observations and increasing by 0.1 for each subsequent observation. The Error Feedback module permits 5 retries. CMA performs Target and Candidates Matching twice: first with a similarity score of 80, then with a user-defined threshold (90 in our experiments).

## 5.2 PERFORMANCE COMPARISON WITH BASELINE METHODS

As shown in Table 1, both CFSA and CMA consistently outperform baselines across heterogeneous EMR schemas. On MIMIC-III, CFSA achieves an F1 of 0.94, surpassing single-prompt baselines (*e.g.*, ICL-PLUQ, 0.749 F1) as well as more complex pipelines. Even under more complex and unseen schemas such as eICU and SICdb (Section 4.1), where baseline F1 scores fall below 0.53 and 0.51, respectively, CFSA maintains high performance (0.93 and 0.81), demonstrating strong generalizability. CMA likewise improves mapping F1 by 0.30, 0.52, and 0.32 on MIMIC-III, eICU, and SICdb over the best competitor, underscoring robust cross-database generalization.

## 5.3 COMPONENT-LEVEL ABLATION OF CFSA AND CMA

Table 2 shows that DB Interaction module (SQL-based Observation + Error Feedback) is the most critical component in CFSA, with its removal causing the largest performance drops across all databases. Schema Guideline also yields consistent gains on all datasets. For CMA, Candidates Matching is indispensable, as disabling it collapses performance to near zero, while Schema Guideline further improves robustness across databases.

**Table 3:** Ablation results of (a) CFSA: Impact of removing Documents and Modules (SQL-based Observation, Error Feedback, SchemaGuideline). (b) CMA: Effect of removing Documents and Modules (Candidate Matching, SchemaGuideline)

| (a) Cohort and Feature Selection | | | | | | |
|---|---|---|---|---|---|---|
| **Method** | MIMIC-III | | eICU | | SICdb | |
| | F1 | Acc. | F1 | Acc. | F1 | Acc. |
| Ours | **0.94**±0.01 | **0.893**±0.01 | **0.929**±0.03 | 0.951±0.03 | **0.814**±0.04 | **0.794**±0.04 |
| Ours w/o Documents | 0.844±0.07 | 0.854±0.06 | 0.917±0.03 | **0.952**±0.03 | 0.748±0.05 | 0.64±0.05 |
| Ours w/o Documents, Modules | 0.443±0.05 | 0.499±0.05 | 0.0±0.0 | 0.0±0.0 | 0.427±0.06 | 0.222±0.03 |
| (b) Code Mapping | | | | | | |
| **Method** | MIMIC-III | | eICU | | SICdb | |
| | F1 | bAcc. | F1 | bAcc. | F1 | bAcc. |
| Ours | **0.516**±0.0 | **0.283**±0.01 | **0.648**±0.05 | **0.336**±0.03 | **0.536**±0.03 | **0.38**±0.02 |
| Ours w/o Documents | 0.336±0.03 | 0.19±0.02 | 0.322±0.03 | 0.208±0.01 | 0.138±0.03 | 0.072±0.02 |
| Ours w/o Documents, Modules | 0.0±0.0 | 0.0±0.0 | 0.07±0.0 | 0.035±0.0 | 0.0±0.0 | 0.0±0.0 |

## 5.4 ROLE OF EXTERNAL KNOWLEDGE

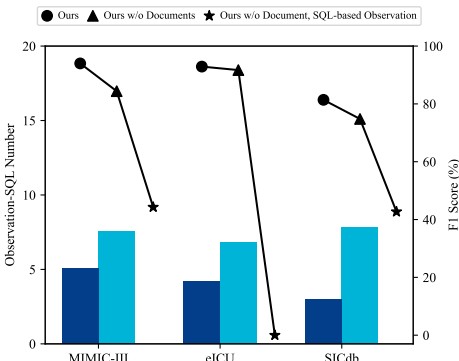

**Figure 3:** Comparison of Observation-SQL Number and F1 Score across EMR databases.

Table 3 show that external knowledge from Documents is essential for both CFSA and CMA. Removing Documents consistently reduces performance, with CMA dropping sharply across all databases. Eliminating both Documents and modules causes near-complete collapse in CMA and substantial declines in CFSA, notably in eICU. Figure 3 further shows that the number of observation SQL queries rises without Documents, indicating a compensatory response to the lack of knowledge. Moreover, when all modules are absent after the missing Documents, performance degradation becomes critical, notably in eICU, highlighting the essential role of integrated components and external knowledge.

**Table 4:** Performance of CFSA and CMA on SICdb with different backbone LLMs.

| Metric | Qwen2.5-72B | Llama-3.1-70B | Claude-3.5-haiku | Claude-3.7-Sonnet | Claude-3.5-Sonnet |
|---|---|---|---|---|---|
| CFSA F1 | 0.22±0.05 | 0.18±0.04 | 0.74±0.05 | 0.80±0.05 | **0.81**±0.04 |
| CFSA Acc | 0.20±0.04 | 0.17±0.04 | 0.69±0.04 | 0.77±0.03 | **0.79**±0.04 |
| CMA F1 | 0.31±0.01 | 0.14±0.02 | 0.44±0.00 | **0.63**±0.02 | 0.54±0.03 |
| CMA bAcc. | 0.16±0.01 | 0.09±0.01 | 0.35±0.00 | **0.39**±0.01 | 0.38±0.02 |

## 5.5 COMPARISON ACROSS VARIOUS BACKBONE MODELS

Table 4 compares CFSA and CMA on SICdb using different LLM backbones. Claude-3.5-Sonnet and Claude-3.7-Sonnet (Anthropic, 2025) achieve the strongest results, with CFSA F1 0.81 and CMA F1 0.63, demonstrating the robustness of the Claude family for EMR preprocessing. In contrast, open-source models Qwen2.5-72B (Yang et al., 2024) and Llama-3.1-70B (Grattafiori et al., 2024) perform poorly, with CFSA F1 0.22 and CMA F1 0.31. Meanwhile, Claude-3.5-haiku (Anthropic, 2024b) offers a computationally efficient alternative, delivering competitive performance with CFSA F1 0.74 and CMA F1 0.44 despite its smaller size.

## 6 CONCLUSION

We present EMR-AGENT, an innovative framework for automated EMR preprocessing using LLM-based agents to replace manual, rule-based methods. Through dynamic database interactions, CFSA and CMA demonstrated robust performance across diverse EMR databases. Although direct comparisons are limited due to the novelty of our approach, evaluations against adapted methods and component-level ablation studies confirmed the effectiveness of our framework in handling heterogeneous data environments. EMR-AGENT suggests a new paradigm for moving beyond rule-based preprocessing, enabling more flexible and scalable EMR data harmonization. An additional discussion covering the limitations and broader impacts of EMR-AGENT is provided in Appendix E.

**Ethics Statement**    This study uses only publicly available and de-identified EMR datasets (MIMIC-III, eICU, and SICdb). All experiments were conducted in compliance with the PhysioNet Data Use Agreement. According to PhysioNet's official guidance, using de-identified data with LLM APIs does not violate the DUA, provided that the API provider does not retain or train on the data. Anthropic's terms of service explicitly state that API inputs are not used for model training or data retention, ensuring our use of Claude-3.5-Sonnet complies with the PhysioNet DUA. We do not manage or provide access to the datasets. The purpose of EMR-AGENT is strictly research-oriented: to advance reproducible and scalable methods for EMR preprocessing. We emphasize that any future clinical deployment would require additional regulatory approval and expert validation to ensure patient safety and fairness.

**Reproducibility Statement**    We took multiple steps to ensure reproducibility. The architecture of EMR-AGENT (CFSA and CMA), training setup, evaluation protocols, and ablation designs are described in detail in the main text and Appendices C to E. Prior to use, one must complete the required credentialing process to access PhysioNet's open datasets. The PreCISE-EMR benchmark provides standardized PostgreSQL database setups and evaluation settings for MIMIC-III, eICU, and SICdb, ensuring consistent execution across environments. The source code of the complete EMR-AGENT framework and PreCISE-EMR benchmark codebases will be released publicly upon acceptance, enabling independent verification and extension of our results.

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

# A APPENDIX A. DETAILS OF *PreCISE-EMR*: PREPROCESSING BENCHMARK

## A.1 EMR DATABASE DESCRIPTION

Table A.1 summarizes the EMR databases included in our *PreCISE-EMR* benchmark. We use MIMIC-III (v1.4), eICU (v2.0), and SICdb (v1.0.8), ensuring compatibility with widely adopted open-source EMR database setup protocols (see Section 4.1).

**Table A.1:** Types and purposes of datasets used in study.

| Dataset | Version | Published | Use | Purpose |
|---|---|---|---|---|
| MIMIC-III (Johnson et al., 2016) | 1.4 | May, 2016 | ✓ | EMR database environment |
| eICU (Pollard et al., 2019) | 2.0 | Apr, 2019 | ✓ | EMR database environment |
| SICdb (Rodemund et al., 2023) | 1.0.8 | Sep, 2024 | ✓ | EMR database environment |
| HiRID (Faltys et al., 2021) | 1.1.1 | Feb, 2021 | △ | Reference for the feature list |

Note that since the feature names in the HiRID (v1.1.1) (Faltys et al., 2021) dataset are defined with standard terminology, it was used as a reference when selecting the mapping code feature list and was excluded from the EMR database environment. For MIMIC-III and eICU, we used official source code[1][2].

Our benchmark, *PreCISE-EMR*, provides hard-coded preprocessing code for two evaluation tasks: (1) Cohort and Feature Selection and (2) Code Mapping.

## A.2 COHORT AND FEATURE SELECTION

### A.2.1 BENCHMARK CONSTRUCTION

For the evaluation of cohort and feature selection, we release a hard-coded benchmark that allows users to specify cohort and feature selection variables. The benchmark enables users to control commonly used inclusion and exclusion criteria, including: 1) age, 2) gender, 3) missing discharge information, 4) minimum ICU stay duration, 5) exclusion of patients with multiple ICU stays, 6) missing gender information, and 7) minimum number of clinical records. These criteria are referenced from well-established studies (Harutyunyan et al., 2019; Sheikhalishahi et al., 2020; van de Water et al., 2024; Wornow et al., 2023). To ensure reliability, we validate our benchmark code using the same cohort criteria as prior benchmarks (Harutyunyan et al., 2019; Sheikhalishahi et al., 2020), confirming that our code extracts identical patient lists under identical criteria (see Figs. A.1 and A.2).

### A.2.2 EVALUATION SET FOR COHORT AND FEATURE SELECTION

Using the released benchmark code (Appendix A.2.1), we construct evaluation sets with natural language inputs that specify (a) user-defined *inclusion and exclusion criteria* (for cohort selection) and (b) user-requested *features* (for feature selection), as summarized in Table A.2 following the column of *Cohort Selection (CS) and Feature Selction (FS)*.

For each evaluation set, the agent must (i) identify the correct cohort (ICU Stays list), with the corresponding patient list reported as **ICU Stays** for each database, and (ii) extract the requested features for these patients in the requested format from *Feature Selection*. Cohort selection accuracy is evaluated by comparing the predicted ICU stay IDs to the gold-standard IDs using the F1-score. Feature Selection accuracy is measured by the correctness of extracted values for the requested features for the patients ICU stays, as shown in Table A.2.

Note that evaluation sets 5, 6, and 7 include *(CMA output)*, indicating that mapping codes are provided. For the cohort and feature selection tasks, ground-truth mapping codes are used, as the performance

---

[1] https://github.com/MIT-LCP/mimic-code/tree/main/mimic-iii/buildmimic/postgres

[2] https://github.com/MIT-LCP/eicu-code/tree/main/build-db/postgres

[3] MIMIC-III: https://github.com/YerevaNN/mimic3-benchmarks/tree/v1.0.0-alpha

[4] eICU: https://github.com/mostafaalishahi/eICU_Benchmark

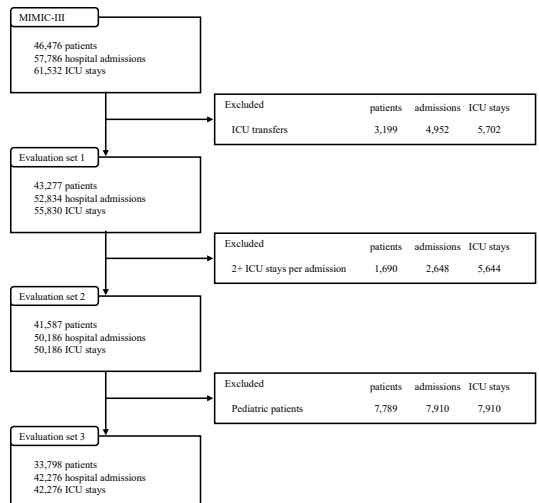

**Figure A.1:** A flowchart for comparison of MIMIC-III benchmark[3] as a reliability evaluation.

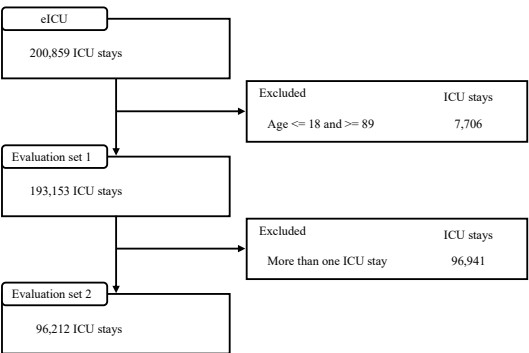

**Figure A.2:** A flowchart for comparison in eICU benchmark[4] as a reliability evaluation.

of the code mapping task is evaluated separately. Each evaluation set was run 10 times (for a total of 70 scores), and the final results were obtained by averaging across trials.

**Table A.2:** User-requested Inclusion and Exclusion criteria (Cohort Selection) applied for Harmonizability evaluation and User-Requested Feature Format (Feature Selection). The base cohorts corresponding to ICU stays in MIMIC-III, eICU, and SICdb are 61,532, 200,859, and 21,932, respectively. (CMA output) represents the prediction output from (Code Mapping Agent).

| Evaluation set | Cohort Selection (CS) and Feature Selection (FS) | ICU Stays (N) | | |
|---|---|---|---|---|
| | | MIMICIII | eICU | SICDb |
| 1 | **CS**: Include only Age 19 to 29 and Include only Male and Exclude ICU stays with missing discharge time
**FS**: ICU-stay id, gender (Male/Female/Unknown), age (integer), length of stay (hours, rounded to 4 decimals in float format) | 1,303 | 4,797 | 428 |
| 2 | **CS**: Include only Age 61 to 69 and Include only Female and Include only ICU stays with at least 30 hours duration
**FS**: ICU-stay id, gender (Male/Female/Unknown), age (integer), mortality status (Dead/Alive/Unknown) | 2,960 | 10,257 | 519 |
| 3 | **CS**: Include only Age 70 to 89 and Include only Male and Exclude stay with multiple ICU stays
**FS**: ICU-stay id, gender (Male/Female/Unknown), age (integer), mortality status (Dead/Alive/Unknown) | 5,603 | 18,387 | 4,965 |
| 4 | **CS**: Include only ICU stays from patients aged 20 to 30 and Exclude patient with missing gender information and Include both Female and Male patients
**FS**: ICU-stay id, gender (Male/Female/Unknown), age (integer), mortality status (Dead/Alive/Unknown) | 2,326 | 9,705 | 1,158 |
| 5 | **CS**: Include only ICU stays from patients aged 40 to 55 and Include ICU stays which contains at least one clinical recrod of 'Hemoglobin [Mass/volume] in Arterial blood **(CMA output)**'
**FS**: ICU-stay id, gender (Male/Female/Unknown), age (integer), mortality status (Dead/Alive/Unknown) | 10,748 | 36,094 | 4,911 |
| 6 | **CS**: Include only ICU stays from patients aged 19 to 30 and Include only Male patients and include stays which contains at least 15 clinical recrod of 'Bicarbonate [Moles/volume] in Arterial blood**(CMA output)**'
**FS**: ICU-stay id, gender (Male/Female/Unknown), age (integer), mortality status (Dead/Alive/Unknown) | 339 | 470 | 206 |
| 7 | **CS**: Include only ICU stays from patients aged 55 to 70 and include ICU stays which contains at least one clinical recrod of 'Lactate [Mass/volume] in Arterial blood**(CMA output)**' or 'Methemoglobin/Hemoglobin.total in Arterial blood**(CMA output)**'
**FS**: ICU-stay id, gender (Male/Female/Unknown), age (integer), mortality status (Dead/Alive/Unknown) | 10,574 | 27,915 | 11,666 |

## A.3 CODE MAPPING

### A.3.1 CODE MAPPING CONSTRUCTION

As described in Section 4.2, we collaborate with a team of five clinical experts (see Fig. A.3) to create code mapping dictionaries for each of the three EMR databases: MIMIC-III, eICU, and SICdb.

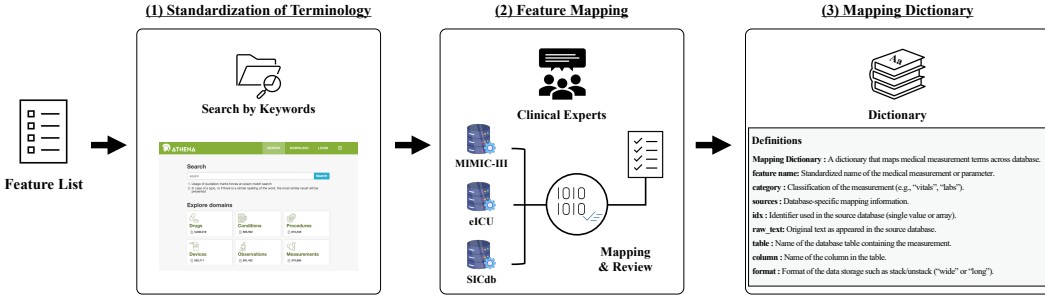

**Figure A.3:** Illustration of the feature mapping procedure.

### A.3.2 EVALUATION SET FOR CODE MAPPING

Our benchmark *PreCISE-EMR* provides an input set of 56 standardized features, referenced from OHDSI (ATHENA, 2023) and listed in Table A.3. Because a single feature can be represented by multiple codes or names, the total number of distinct codes corresponding to these 56 features is 126 in MIMIC-III, 53 in eICU, and 87 in SICdb. These counts exclude cases where a requested feature does not exist in a given database. As shown in Table A.3, some features are absent in certain databases, resulting in true negatives or false positives during evaluation.

For mapping codes stored as columns, the prediction must include both the table name and column name (*e.g.*, `vitalperiodic.temperature`, `vitalperiodic.systemicsystolic`). For codes stored as rows, the prediction must include both the code number and feature name (*e.g.*, (656, Glukose (BGA)), (348, Glukose (ZL))) for MIMIC-III and SICdb. In eICU, where code numbers are not available, only the feature name is used for code mapping evaluation.

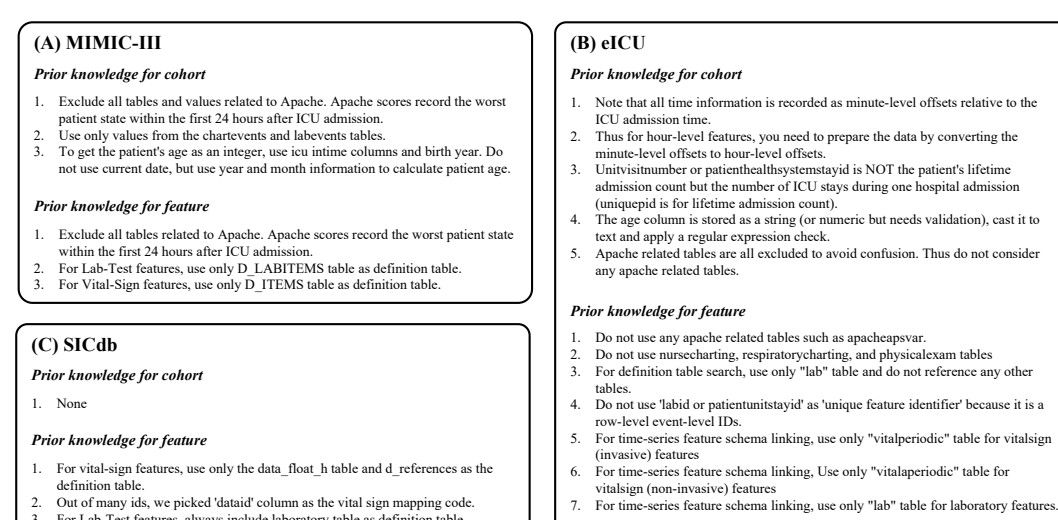

**(A) MIMIC-III**

*Prior knowledge for cohort*

1. Exclude all tables and values related to Apache. Apache scores record the worst patient state within the first 24 hours after ICU admission.
2. Use only values from the chartevents and labevents tables.
3. To get the patient's age as an integer, use icu intime columns and birth year. Do not use current date, but use year and month information to calculate patient age.

*Prior knowledge for feature*

1. Exclude all tables related to Apache. Apache scores record the worst patient state within the first 24 hours after ICU admission.
2. For Lab-Test features, use only D_LABITEMS table as definition table.
3. For Vital-Sign features, use only D_ITEMS table as definition table.

**(C) SICdb**

*Prior knowledge for cohort*

1. None

*Prior knowledge for feature*

1. For vital-sign features, use only the data_float_h table and d_references as the definition table.
2. Out of many ids, we picked 'dataid' column as the vital sign mapping code.
3. For Lab-Test features, always include laboratory table as definition table.
4. Out of many ids, we picked 'laboratoryid' column as the Lab-Test mapping code.

**(B) eICU**

*Prior knowledge for cohort*

1. Note that all time information is recorded as minute-level offsets relative to the ICU admission time.
2. Thus for hour-level features, you need to prepare the data by converting the minute-level offsets to hour-level offsets.
3. Unitvisitnumber or patienthealthsystemstayid is NOT the patient's lifetime admission count but the number of ICU stays during one hospital admission (uniquepid is for lifetime admission count).
4. The age column is stored as a string (or numeric but needs validation), cast it to text and apply a regular expression check.
5. Apache related tables are all excluded to avoid confusion. Thus do not consider any apache related tables.

*Prior knowledge for feature*

1. Do not use any apache related tables such as apacheapsvar.
2. Do not use nursecharting, respiratorycharting, and physicalexam tables
3. For definition table search, use only "lab" table and do not reference any other tables.
4. Do not use 'labid or patientunitstayid' as 'unique feature identifier' because it is a row-level event-level IDs.
5. For time-series feature schema linking, use only "vitalperiodic" table for vitalsign (invasive) features
6. For time-series feature schema linking, Use only "vitalaperiodic" table for vitalsign (non-invasive) features
7. For time-series feature schema linking, use only "lab" table for laboratory features.

**Figure A.4:** Evaluation memos used as a concise note highlighting dataset guideline identified by clinical experts.

**Table A.3:** Feature list used for feature mapping in the framework evaluation set. We explored features using the Observation Source Table in the HiRID dataset (Faltys et al., 2021) defined with standard terminology as a reference, and added features that are commonly used in laboratory tests but not included in HiRID. The newly added features were mapped to standard terminology in Athena OHDSI. Additionally, we limited features to vital signs and laboratory tests, and finally selected features that exist in at least one of the three datasets, resulting in a total of 56 features.

| Feature | MIMIC-III | eICU | SICdb |
|---|---|---|---|
| Core body temperature | ✓ | ✓ | ✓ |
| Heart rate | ✓ | ✓ | ✓ |
| Invasive diastolic arterial pressure | ✓ | ✓ | ✓ |
| Invasive mean arterial pressure | ✓ | ✓ | ✓ |
| Invasive systolic arterial pressure | ✓ | ✓ | ✓ |
| Non-invasive diastolic arterial pressure | ✓ | ✓ | ✓ |
| Non-invasive mean arterial pressure | ✓ | ✓ | ✓ |
| Non-invasive systolic arterial pressure | ✓ | ✓ | ✓ |
| Respiratory rate | ✓ | ✓ | ✓ |
| Alanine aminotransferase [Enzymatic activity/volume] in Serum or Plasma | ✓ | ✓ | ✓ |
| Albumin [Mass/volume] in Serum or Plasma | ✓ | ✓ | ✓ |
| Alkaline phosphatase [Enzymatic activity/volume] in Blood | ✓ | ✓ | ✓ |
| aPTT in Blood by Coagulation assay | ✗ | ✓ | ✓ |
| Aspartate aminotransferase [Enzymatic activity/volume] in Serum or Plasma | ✓ | ✓ | ✓ |
| Band form neutrophils/100 leukocytes in Blood | ✓ | ✗ | ✓ |
| Base excess in Arterial blood by calculation | ✓ | ✓ | ✓ |
| Bicarbonate [Moles/volume] in Arterial blood | ✓ | ✓ | ✓ |
| Bilirubin.direct [Mass/volume] in Serum or Plasma | ✓ | ✓ | ✓ |
| Bilirubin.total [Moles/volume] in Serum or Plasma | ✓ | ✓ | ✓ |
| C reactive protein [Mass/volume] in Serum or Plasma | ✓ | ✓ | ✓ |
| Calcium [Moles/volume] in Blood | ✓ | ✓ | ✓ |
| Calcium.ionized [Moles/volume] in Blood | ✓ | ✓ | ✓ |
| Carbon dioxide [Partial pressure] in Arterial blood | ✓ | ✓ | ✓ |
| Chloride [Moles/volume] in Blood | ✓ | ✓ | ✓ |
| Cholesterol [Mass/volume] in Serum or Plasma | ✓ | ✓ | ✓ |
| Creatine kinase [Mass/volume] in Blood | ✓ | ✓ | ✓ |
| Creatine kinase.MB [Mass/volume] in Blood | ✓ | ✓ | ✗ |
| Creatine kinase.MB [Mass/volume] in Serum or Plasma | ✗ | ✗ | ✓ |
| Creatinine [Moles/volume] in Blood | ✓ | ✓ | ✓ |
| Fibrinogen [Mass/volume] in Platelet poor plasma by Coagulation assay | ✓ | ✓ | ✓ |
| Glucose [Moles/volume] in Serum or Plasma | ✓ | ✓ | ✓ |
| Hematocrit [Volume Fraction] of Blood | ✓ | ✓ | ✓ |
| Hemoglobin [Mass/volume] in Arterial blood | ✓ | ✓ | ✓ |
| INR in Blood by Coagulation assay | ✓ | ✓ | ✗ |
| Lactate [Mass/volume] in Arterial blood | ✓ | ✓ | ✓ |
| Leukocytes [#/volume] in Blood | ✓ | ✗ | ✓ |
| Lymphocytes [#/volume] in Blood | ✓ | ✓ | ✓ |
| Magnesium [Moles/volume] in Blood | ✓ | ✓ | ✓ |
| MCH - Mean corpuscular haemoglobin | ✓ | ✓ | ✓ |
| MCHC [Mass/volume] | ✓ | ✓ | ✓ |
| MCV [Entitic volume] | ✓ | ✓ | ✓ |
| Methemoglobin/Hemoglobin.total in Arterial blood | ✓ | ✓ | ✓ |
| Neutrophils/100 leukocytes in Blood | ✓ | ✓ | ✓ |
| Oxygen [Partial pressure] in Arterial blood | ✓ | ✗ | ✓ |
| Oxygen measurement, partial pressure, arterial | ✓ | ✓ | ✓ |
| Oxygen saturation in Arterial blood | ✓ | ✓ | ✓ |
| Partial thromboplastin time ratio | ✓ | ✓ | ✗ |
| pH of Arterial blood | ✓ | ✓ | ✓ |
| Phosphate [Moles/volume] in Blood | ✓ | ✓ | ✓ |
| Platelets [#/volume] in Blood | ✓ | ✓ | ✓ |
| Potassium [Moles/volume] in Blood | ✓ | ✓ | ✓ |
| Sodium [Moles/volume] in Blood | ✓ | ✓ | ✓ |
| Troponin I measurement | ✓ | ✓ | ✓ |
| Troponin T.cardiac [Mass/volume] in Serum or Plasma | ✓ | ✓ | ✓ |
| Urea [Moles/volume] in Venous blood | ✗ | ✗ | ✓ |
| Urea nitrogen [Mass/volume] in Serum or Plasma | ✓ | ✓ | ✗ |

## A.4 EVALUATION MEMO

For both **Cohort and Feature Selection Evaluation** and **Code Mapping Evaluation**, our benchmark *PreCISE-EMR* includes evaluation memos specifically for each EMR database. Each memo details the rules followed by clinical experts during the construction of the evaluation set. For both CFSA and CMA tasks, there is no database-specific information in the prompts apart from this evaluation memo, database metadata (including EMR database manual), and schema information. These memos were created prior the evaluation set construction and are shown in Fig. A.4.

## B  APPENDIX B. CASE STUDY OF AGENT REASONING

This appendix provides a concise walkthrough illustrating how EMR-AGENT performs Cohort/Feature Selection (CFSA) and Code Mapping (CMA). Each figure presents the key steps taken by each agent.

### B.1  COHORT AND FEATURE SELECTION AGENT (CFSA) CASE STUDY

In Figure B.1, the agent begins by interpreting the user instructions together with the database metadata. It then performs schema linking (mapping the instruction to relevant tables and columns in SICdb) and retrieves the corresponding schema guidelines (e.g., what information is missing to fulfill the user request, the meaning of each column, etc.). In the Figure B.1, the agent identifies key elements such as the **sex column**, **d_reference table**, and **caseid column**.

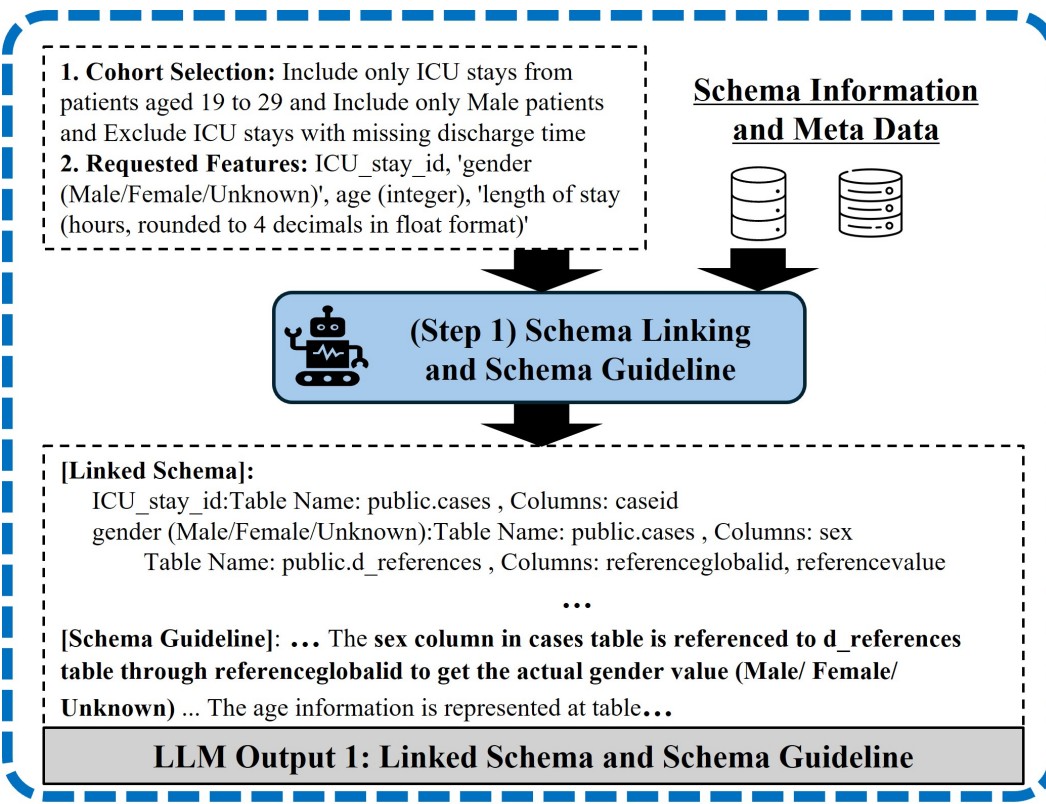

**Figure B.1: Step 1: Schema Linking and Schema Guideline.** The agent identifies tables/columns for ICU_stay_id, gender, age, and length of stay, and constructs an initial schema guideline.

In Figure B.2, after linking the relevant columns, the agent evaluates whether additional information from the EMR is required. Because the gender-code pairing is not yet known, the agent decides that further observation is necessary and issues a SQL-observation query to inspect the actual stored values. Through this inspection, the agent discovers that SICdb encodes gender using numeric reference IDs (735 for male and 736 for female). The agent also figures out that the gender code must be resolved via a join with **d_references**. Recognizing this as essential schema information, the agent updates the schema guideline accordingly.

In Figure B.3, With the mappings and filters resolved, the agent composes the final SQL query satisfying all constraints: age between 19–29, male-only, and non-null LOS, with LOS converted from seconds to hours. If the SQL execution produced a syntax error, it would feedback to SQL-generation step with the error message. If the SQL exectution produced a schema mismatch error

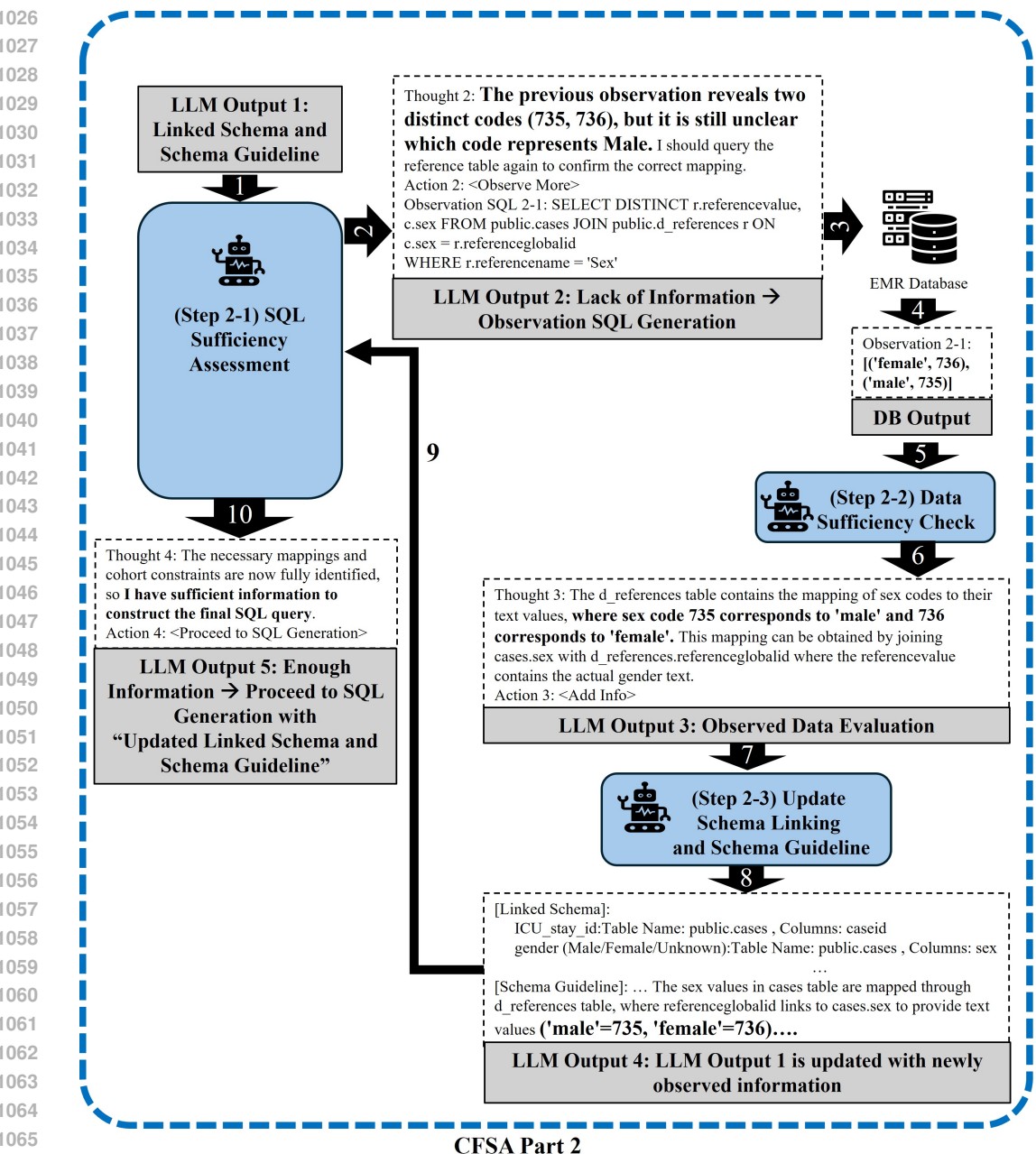

**CFSA Part 2**

**Figure B.2: Step 2: SQL Observation.** Observation queries reveal the true gender mapping: 735 = Male, 736 = Female. The agent updates its schema guideline and confirms sufficiency for SQL generation.

(e.g., missing columns, invalid types), it would feedback to Schema Linking and Schema Guideline step with the error message.

## B.2 CODE MAPPING AGENT (CMA) CASE STUDY

In Figure B.4, the agent begins by examining **d_labitems** and **labevents**, together with the MIMIC-III metadata, to understand how laboratory measurements are defined and stored in MIMIC-III. This step establishes the structural context necessary to search for the requested feature, *Hemoglobin [Mass/Volume] in Arterial Blood,* before attempting any candidate matching or Feature Locating.

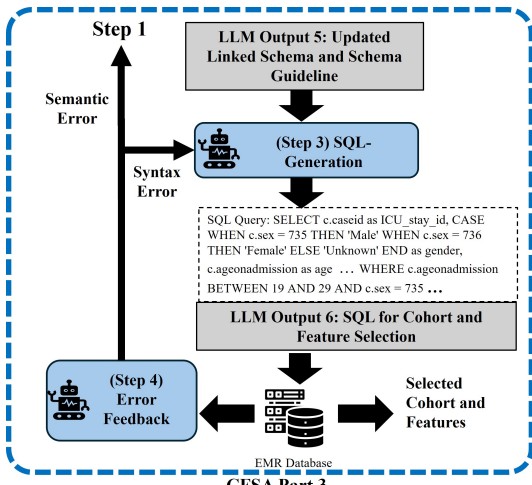

**Figure B.3: Step 3: Final SQL Generation.** The agent produces a correct SQL query with all filters applied and LOS properly converted and rounded.

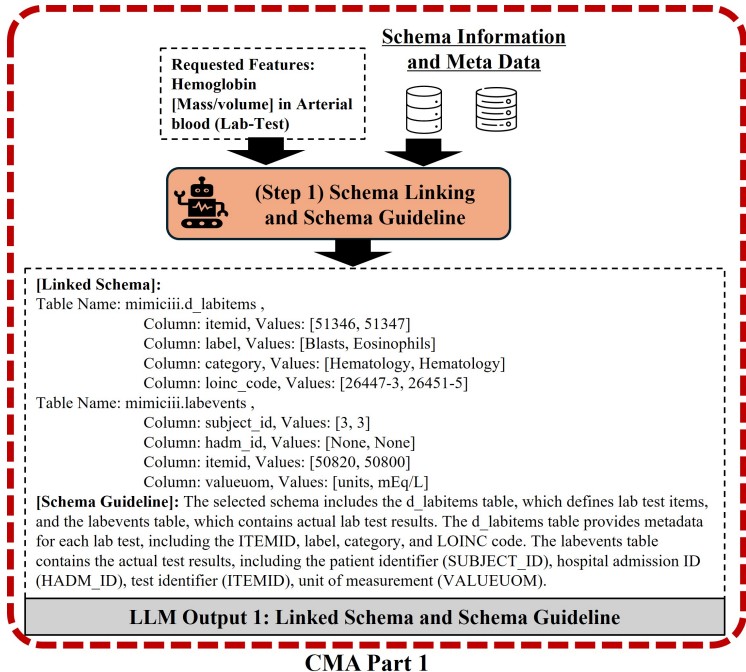

**Figure B.4: Step 1: Schema Linking for Laboratory Measurements.** The agent selects and inspects **d_labitems** and **labevents**, including item labels, units, fluid types, categories, and LOINC codes, to understand how lab tests are represented in MIMIC-III. This schema-level analysis provides the foundation for locating the user-requested feature *Hemoglobin [Mass/Volume] in Arterial Blood*.

In Figure B.5, after the initial schema linking, the agent checks whether the target feature, *Hemoglobin [Mass/volume] in Arterial blood*, appears literally in any column names. Because neither the full name nor any similar name matches the target concept, the agent concludes that literal matching is not feasible. Based on this assessment, it proceeds to the Candidate Matching stage to search for the target item.

In Figure B.6, the agent performs a controlled SQL query to retrieve all candidate laboratory items and then applies LLM-based semantic matching over the resulting metadata. Rather than relying

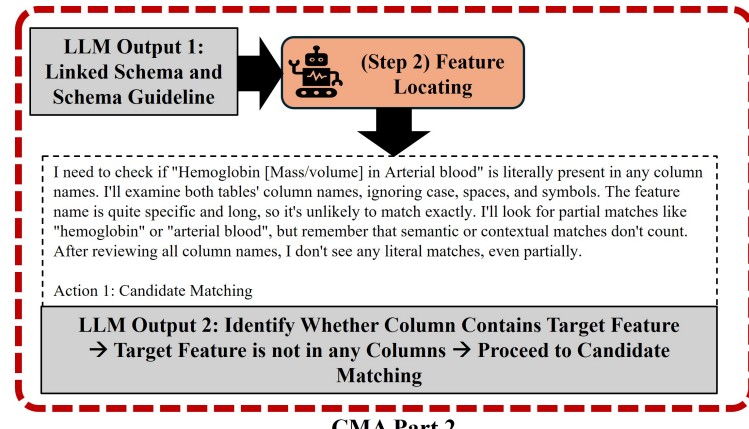

CMA Part 2

**Figure B.5: Step 2: Feature Locating.** The agent verifies that the target concept does not appear as a literal column name in the schema. After confirming the absence of any direct lexical match, the agent transitions to semantic candidate listing, where item-level metadata (e.g., labels, fluids, categories) is examined to identify potential Hemoglobin-related codes.

on raw lexical similarity, the agent compares item-level attributes such as code number, label, unit type, and category to determine which entries correspond to the intended Hemoglobin measurement. Through this semantic reasoning process, the agent narrows the set of Hemoglobin-related itemids candidates.

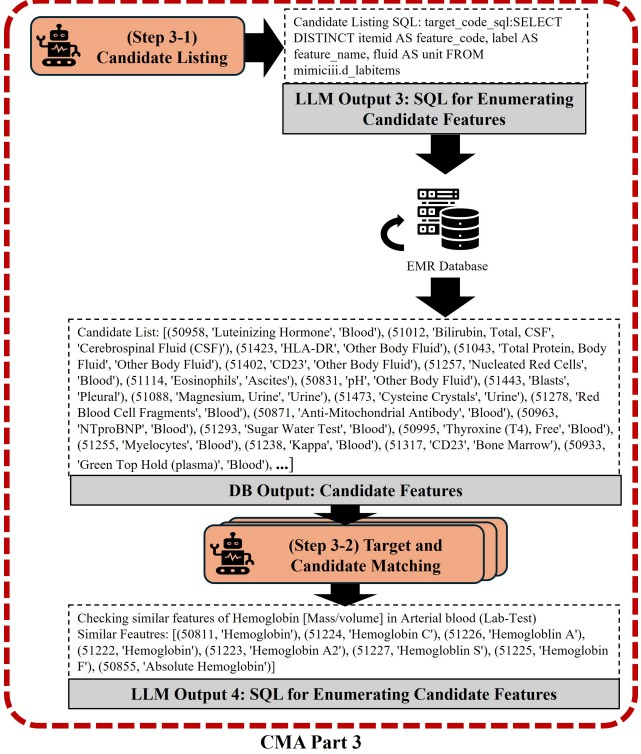

CMA Part 3

**Figure B.6: Step 3: Candidate Matching.** From retrieved candidates, the agent selects multiple Hemoglobin-related itemids through semantic reasoning over item labels and metadata.

# C    APPENDIX C. COST-PERFORMANCE COMPARISON ACROSS BASELINES

We report the average number of LLM calls, runtime, and prediction performance for all baselines across the three EMR datasets. To enable a unified comparison, we present the prediction performance results for Cohort and Feature Selection (CFS) and Code Mapping (CM) side by side.

Tables C.1 and C.2 show that ICL (PLUQ) is a single-shot approach, and SeqSQL and DinSQL rely on fixed call counts imposed by their rigid decomposition. In contrast, both ReAct and EMR-AGENT employ interactive reasoning, resulting in a variable number of LLM calls.

## C.1    COHORT AND FEATURE SELECTION (CFS) COST-PERFORMANCE COMPARISON

**Table C.1: Cohort and Feature Selection (CFS):** LLM call count, runtime (second), and F1 score across MIMIC-III, eICU, and SICdb.

| Model | MIMIC-III | | | eICU | | | SICdb | | |
|---|---|---|---|---|---|---|---|---|---|
| | Calls | Runtime | F1 | Calls | Runtime | F1 | Calls | Runtime | F1 |
| **Ours (CFSA)** | 9.98 | 115.28 | **0.94** | 10.31 | 124.84 | **0.929** | 7.02 | 69.80 | **0.814** |
| ICL (PLUQ) | 1 | 15.55 | 0.749 | 1 | 16.88 | 0.132 | 1 | 10.72 | 0.407 |
| ICL (SeqSQL) | 2.85 | 30.41 | 0.040 | 2.85 | 27.58 | 0.000 | 2.85 | 20.28 | 0.040 |
| DinSQL | 4 | 42.72 | 0.726 | 4 | 38.42 | 0.000 | 4 | 38.30 | 0.071 |
| ReAct | 11.17 | 182.14 | 0.308 | 15 | 266.54 | 0.524 | 10.46 | 72.86 | 0.503 |

**Cost–Performance Analysis for CFS Across Baselines (Table C.1).** EMR-AGENT is more cost-efficient than ReAct: although ReAct makes more LLM calls and has longer runtime across all datasets, it consistently achieves substantially lower F1 accuracy. This indicates that EMR-AGENT's gains are not due to calling the model more frequently. Compared with lightweight baselines such as PLUQ, SeqSQL, and DinSQL, EMR-AGENT uses a moderate number of additional calls but yields large improvements in accuracy. These added calls correspond to *SQL-Observation* and *Error Feedback* steps. Table 2 in the main paper shows that removing these steps causes severe accuracy degradation, confirming that the extra calls are necessary for resolving schema ambiguity, not inefficient recursion.

## C.2    CODE MAPPING (CM) COST-PERFORMANCE COMPARISON

**Table C.2: Code Mapping (CM):** LLM call count, runtime (second), and F1 score across MIMIC-III, eICU, and SICdb.

| Model | MIMIC-III | | | eICU | | | SICdb | | |
|---|---|---|---|---|---|---|---|---|---|
| | Calls | Runtime | F1 | Calls | Runtime | F1 | Calls | Runtime | F1 |
| **Ours (CMA)** | 17.48 | 72.97 | **0.516** | 16.26 | 53.02 | **0.648** | 18.46 | 73.11 | **0.536** |
| ICL (PLUQ) | 1 | 12.92 | 0.022 | 1 | 9.88 | 0.125 | 1 | 8.33 | 0.119 |
| ReAct | 9.17 | 21.80 | 0.214 | 4.47 | 64.82 | 0.067 | 9.17 | 16.84 | 0.218 |

**Cost–Performance Analysis for CM Across Baselines (Table C.2).** Across all datasets, EMR-AGENT achieves substantially higher F1 performance compared to PLUQ and ReAct. While PLUQ has the lowest runtime due to being a single-shot method, its accuracy remains near-zero across datasets. ReAct performs multiple iterative steps but still underperforms both in accuracy compared to EMR-AGENT. Overall, the results show that EMR-AGENT achieves the strongest code-mapping accuracy with a moderate number of LLM calls, representing a favorable cost-performance trade-off relative to existing baselines.

# D    APPENDIX D. QUANTIFYING HUMAN-IN-THE-LOOP EFFORT

This appendix provides a detailed analysis of the human-effort implications of EMR-AGENT for EMR onboarding. This section quantifies how much expert correction remains after automation.

### D.1. HUMAN CORRECTION REQUIREMENTS IN COHORT & FEATURE SELECTION (CFS)

Onboarding a new EMR requires clinicians to interpret schema structures, write SQL logic, and verify extracted feature values. EMR-AGENT automates these components, and we measure how much manual work remains after automation.

Residual human effort in CFS is quantified using strict correctness measures:

- **Cohort Logic Errors:** A cohort clause is marked incorrect if even a single patient differs from the reference output. Any discrepancy indicates a semantic mismatch that requires full clinician re-validation.
- **Feature Logic Errors:** If any extracted value for gender, age, or LOS is incorrect for any patient, the entire feature clause is considered erroneous, as such errors typically reflect incorrect joins, filters, or temporal logic.

These error categories reflect upper bounds on the additional manual debugging required during clinician review. In practice, clinicians must still inspect every generated SQL clause (even those that are fully correct) because EMR outputs cannot be accepted without human verification. However, starting from an agent-generated SQL draft is far more efficient than writing queries from scratch, and clauses that match the reference typically require only brief confirmation. In contrast, any discrepancy triggers substantially more work, including re-examining joins, filters, or temporal logic.

Each database includes 70 CFS tasks, for a total of 210 outputs across MIMIC-III, eICU, and SICdb. All 210 outputs require at least basic review, but logic errors indicate cases where clinicians must move beyond light inspection and perform full debugging. Therefore, reductions in logic errors correspond to meaningful reductions in the depth of human corrective effort.

**Table D.1: Human correction requirements for Cohort & Feature Selection (CFS).** EMR-AGENT produces substantially fewer erroneous SQL clauses than all baselines.

| Method | Cohort Logic Errors | | Feature Logic Errors | |
|---|---|---|---|---|
| | Correct Patients | Erroneous Patients | Correct Features | Erroneous Features |
| **Ours (CFSA)** | **157** | **53** | **108** | **102** |
| ICL (PLUQ) | 47 | 153 | 11 | 199 |
| ICL (SeqSQL) | 4 | 206 | 0 | 210 |
| DinSQL | 15 | 195 | 3 | 207 |
| REACT | 44 | 156 | 27 | 183 |

Across 210 tasks, EMR-AGENT generates 155 logic errors, compared to 336 to 416 errors by baselines. This corresponds to a 54 to 63% reduction in manual debugging effort.

### D.2. HUMAN-LABOR REDUCTION IN CODE MAPPING (CM)

Code Mapping requires extensive manual search effort. We distinguish between low-cost (TP/FP) and high-cost (FN/TN/Remaining) clinician operations. High-cost operations correspond to searching thousands of raw EMR codes, which dominate real-world preprocessing time. Thus, the total number of FN + TN + Remaining concepts represents human labor. "Remaining" denotes unmapped concepts requiring manual search, which are equivalent in cost to FN/TN operations.

We evaluate CM over 56 clinical concept types. Because one concept can correspond to multiple raw codes, each EMR contains different numbers of raw measurement codes (52 in eICU, 115 in MIMIC-III, and 132 in SICdb), and some concepts are absent in each EMR (maximal TN = 5, 3, and 4, respectively). For robustness, the CM evaluation is repeated three times (three independent runs) using the same set of clinical concepts, yielding a total of 933 concept-level mapping checks across the three EMRs: $3 \times (52 + 115 + 132 + 5 + 3 + 4)$.

EMR-AGENT reduces high-cost operations from 933 to 399, a 57% reduction in the most labor-intensive part of EMR onboarding. Baselines reduce only 5 to 17%. This metric requires no assumptions about annotation time, but it reflects the intrinsic asymmetry that FN/TN searches are

**Table D.2: Human-labor reduction in Code Mapping (CM).** High-cost operations correspond directly to manual search burden.

| Method | FN | TN | Remaining[†] | High-Cost Ops | Reduction vs. Human |
|---|---|---|---|---|---|
| **Human (baseline)** | 0 | 0 | 933 | 933 | – |
| **Ours (CMA)** | 340 | 26 | 33 | **399** | **57%** ↓ |
| ICL (PLUQ) | 792 | 17 | 82 | 891 | 5% ↓ |
| REACT | 652 | 31 | 87 | 770 | 17% ↓ |

[†]Remaining = 933 − TP − FN − TN.

vastly more expensive than TP/FP verification. To avoid trivial over-predicting, this evaluation metric must be interpreted with predictive accuracy (Table 1), preventing degenerate solutions.

### D.3. OVERALL IMPACT ON HUMAN-IN-THE-LOOP WORKLOAD

Across both CFS and CM, EMR-AGENT substantially lowers the amount of manual correction required from clinicians. While expert verification remains essential for safety, the automated pipeline meaningfully reduces the scope of human intervention needed for EMR preprocessing.

## E    APPENDIX E. LIMITATIONS AND BROADER IMPLICATIONS

**Limitation**    EMR-AGENT is not designed to fully replace human expertise in EMR preprocessing. While it significantly automates data extraction, it remains a supportive tool that requires validation by qualified professionals to ensure accuracy. Unlike hard-coded pipelines that are specifically tailored to individual datasets and can achieve near-perfect accuracy, EMR-AGENT may not consistently reach this level of precision. As a result, data extracted by the agent may require further validation before being used in time-series tabular model training.

Our evaluation focuses on cohort selection scenarios designed to ensure clear, reproducible ground truth across the three ICU datasets. More complex clinical conditions (e.g., sepsis, ventilation) were not included because their definitions vary across institutions and lack dataset-wide consensus standards. We plan to extend the benchmark to compound and temporal cohort definitions in future work.

A similar limitation applies to our code mapping evaluation. Our current setup focuses on common laboratory tests, which represent one of the few clinical domains where clear and reproducible reference standards exist. More complex concepts such as medications and ventilation parameters would require integration of broader reference terminologies (e.g., RxNorm, SNOMED-CT), dose/route normalization, handling of time-varying concepts, and development of appropriate evaluation frameworks. While EMR-AGENT's modular architecture is designed to accommodate such extensions, establishing reliable reference standards for these domains remains an open challenge, and we plan to address these limitations in future work.

Furthermore, it is important to acknowledge that EMR-AGENT may carry forward existing biases present in the raw EMR data. These biases, which often stem from historical healthcare inequities and varying data collection practices across different demographic groups, may appear in various forms, such as disparities in demographics, diagnoses, or treatments embedded within the EMR databases. The automated extraction process, while efficient, does not inherently address or mitigate these systematic biases, which can subsequently influence downstream machine learning models. Researchers utilizing EMR-AGENT should be aware of these limitations and implement appropriate strategies for bias detection and fairness assessment in their analyses.

Nevertheless, this agent-based approach introduces a scalable and adaptive paradigm with promising potential for future improvements.

**Broader Impacts**    EMR-AGENT has the potential to reduce the manual workload for clinical experts in managing complex EMR data. However, given the sensitive nature of healthcare information,

its deployment must be accompanied by rigorous validation and ongoing oversight to ensure safety, accuracy, and ethical compliance.

Although EMR-AGENT utilizes large language models for database interactions, its environmental impact remains relatively modest as it operates solely during inference, without the need for training or fine-tuning. Moreover, from the perspective of a broader research community, the framework offers significant efficiency gains. By providing a standardized and automated solution, EMR-AGENT reduces the need for multiple research teams to develop similar preprocessing pipelines independently, leading to more resource utilization across the community. The framework's reusability and scalability further distribute this computational cost across multiple studies and datasets, thereby promoting more standardized and reproducible EMR research practices.

This work is expected to inspire further research in the field and contribute to its advancement, while maintaining a balance between computational efficiency and environmental responsibility.

# F   APPENDIX F. BASELINES PROMPTS

Since each baseline model is not designed for our task, we adapt our prompts from the original ones, preserving each model's structural format. In this section, we present the prompt settings for each baseline.

## F.1   ICL IN PLUQ

Here, the schema information format and prompt style are adopted from PLUQ (Jo et al., 2024). This baseline utilizes the LLM in a single-turn setting only.

---

**Cohort and Feature Selection prompt**

You are given Database information and the Question. Generate the PostgreSQL query for the following question. Note that you should generate 'null' if the question cannot be converted to SQL query given information. Get only one SQL query as plain text. Do not include code delimiters (e.g., "'sql), comments, or any additional text.

[Schema Information]: {Schema_information}
[Evaluation Memo]: {Evaluation_Memo}
[Database Manual]: {Database_Manual}

Q: List all {Feature_Selection} information that satisfy following [Cohort Selection].
[Cohort Selection]: {Cohort_Selection}
Ensure the output in PostgreSQL strictly follows the order and format specified in each () of {Feature_Selection}.
SQL Query:

---

**Code mapping prompt**

You are given a [Database schema] and a [Feature].
Task: Analyze the information provided below and classify the feature into one of the following categories:
<get schema>: Select this if you can find a column whose name literally matches any part of the given [Feature].
<get definition SQL>: Select this if no such column exists, but you can retrieve the corresponding feature information using an SQL query. The query should return the unique feature identifier, feature name, and unit from the definition table related to the [Feature].
<null>: Select this if neither a matching schema nor an SQL definition can be found. Instructions:

- If you choose <get schema>, provide the matching table and column in the format: Table_Name.Column_Name

---

- If you choose <get definition SQL>, provide an SQL query in the format: SELECT unique_feature_identifier, feature_Name, unit FROM dbname.Table_A WHERE ...

[Schema Information]: {Schema_information}
[Evaluation Memo]: {Evaluation_Memo}
[Database Manual]: {Database_Manual}
[Feature] : {Target_Feature}

Output Format:
<classification>
<get schema>, <get mapping SQL>, or <null>
</classification>
<answer>
[get answer for selected classification formation]
</answer>

## F.2 ICL IN SEQSQL

SeqSQL (Ryu et al., 2024) is a sequential generation approach for complex SQL queries by decomposing cohort selection into individual conditions. Each decomposed condition's SQL is generated step-by-step, leveraging the outputs of previous steps to structurally compose the final SQL query. For the Cohort and Feature Selection task, we generate SQL queries corresponding to various conditions and implemented the baseline by combining these conditions using logical conjunctions ("and") as shown in Listing 1. And we utilize all prompt structure from (Ryu et al., 2024) except for 20-shot Examples; Post-processing Detail, SQL-like Rep.Description, Test Question. However, for the Code Mapping task, where the core idea is database search based on a single condition, the use of SeqSQL was unsuitable due to the mismatch in task characteristics, and thus it was not implemented for comparison.

```python
question_all = []

question_information = f"List all {requested_features.strip()}
    information. Ensure the output in PostgreSQL strictly follows the
    order and format specified in each () of {requested_features.strip()
    }."
question_all.append(question_information)

cohort_selection = cohort_selection.split("and")
if isinstance(cohort_selection, list):
    for condition in cohort_selection:
        question_information = f"Retrieve only the cases \"{condition.
            strip()}\""
        question_all.append(question_information)
else:
    question_information = f"Retrieve only the cases \"{cohort_selection
        }\""
    question_all.append(question_information)
```

**Listing 1:** Logic of spliting cohort selection into simple condition in Python

### Cohort and Feature Selection prompt

Get only one PostgreSQL query as plain text. Do not include code delimiters (e.g., "'sql), comments, or any additional text.

[Schema Information]: {Schema_information}
[Evaluation Memo]: {Evaluation_Memo}

[Database Manual]: {Database_Manual}

– Post-processing Detail
Please note that:
1. Questions asking whether a specific number falls within a normal range can be formulated as follows and will be changed through the post_processing process.
NLQ: Had the value of result measured during result been normal?
SQL: SELECT COUNT(*)>0 FROM chartevents WHERE chartevents.icustay_id IN (... ) AND chartevents.valuenum BETWEEN sao2_lower AND sao2_upper
2. Similarly, for questions that require the current time, we will use 'current_time' as a placeholder and adjust it as necessary. For reference, the current time is assumed to be "2105-12-31 23:59:00". Therefore, if there is the expression "this month" means 2105-12.

– SQL-like Rep. Description
PREV_QUERY and PREV_RESULT tokens allow for referencing and reusing the SQL code and results of previous queries in subsequent ones.
The PREV_QUERY token is used to represent the SQL code of the previous query, essentially allowing the new query to build upon it or modify it. SQL queries can also start with the PREV_QUERY token, which enables the duplication and utilization of the previous query in the new one.
The PREV_RESULT token, on the other hand, is used to represent the example of result set from a previous query, rather than the query itself. This is useful when we want to use the results of a previous query directly within a new query.

– TEST_QUESTION
NLQ1:{question_all[0]}
SQL1:

## F.3    DINSQL

DinSQL (Pourreza & Rafiei, 2023) generates SQL queries by selecting the most appropriate schema based on both the database information and the given cohort selection condition. Then it classifies the complexity of the condition and generates SQL by its complex state followed with self-correction mechanism. DinSQL is comparable to our method in its ability to handle complex condition-based SQL generation, making it suitable for comparison in the Cohort and Feature Selection task, it is not appropriate for Code Mapping, which is about database searching for simple, single-condition. Thus DinSQL has been used in Cohort and Feature Selection task.

### F.3.1    COHORT AND FEATURE SELECTION

**Schema linking prompt**

# Find the Schema_links for generating SQL queries for each question based on the [Schema Information], [Evaluation Memo], and [Database Manual].

[Schema Information]: {Schema_information}
[Evaluation Memo]: {Evaluation_Memo}
[Database Manual]: {Database_Manual}

Q: List all {Target_Features} information that satisfy following [Cohort Selection].
[Cohort Selection]: {Cohort_Selection}
Make PostgreSQL follow the order of the provided information, categories, type.
A: Let's think step by step.

**Classification prompt**

# For the given question, classify it as EASY, NON-NESTED, or NESTED based on nested queriesand JOIN.

if need nested queries: predict NESTED
elif need JOIN and don't need nested queries: predict NON-NESTED
elif don't need JOIN and don't need nested queries: predict EASY

Q: List all {Target_Features} information that satisfy following [Cohort Selection].
[Cohort Selection]: {Cohort_Selection}
Make PostgreSQL follow the order of the provided information, categories, type.
Schema_links: {Schema_links}
A: Let's think step by step.

**SQL generation prompt**

**easy_prompt**

# Use the the schema links to generate the SQL queries for each of the questions.

Q: "Find the buildings which have rooms with capacity more than 50."
Schema_links: [classroom.building,classroom.capacity,50]
SQL: SELECT DISTINCT building FROM DB_Name.classroom WHERE capacity > 50
. . .

Q: List all {Target_Features} information that satisfy following [Cohort Selection].
[Cohort Selection]: {Cohort_Selection}
Make PostgreSQL follow the order of the provided information, categories, type.
Schema_links: {Schema_links}
SQL:

**medium_prompt**

# Use the the schema links and Intermediate_representation to generate the SQL queries for each of the questions.

Q: "Find the total budgets of the Marketing or Finance department."
Schema_links: [department.budget,department.dept_Name,Marketing,Finance]
A: Let's think step by step. For creating the SQL for the given question, we need to join these tables = []. First, create an intermediate representation, then use it to construct the SQL query.
Intermediate_representation: select sum(department.budget) from department where department.dept_Name = "Marketing" or department.dept_Name = "Finance"
SQL: SELECT sum(budget) FROM DB_Name.department WHERE dept_Name = 'Marketing' OR dept_Name = 'Finance'
. . .

Q: List all {Target_Features} information that satisfy following [Cohort Selection].
[Cohort Selection]: {Cohort_Selection}
Make PostgreSQL follow the order of the provided information, categories, type.
Selected_Schema: {Selected_Schema}
A: Let's think step by step.

**hard_prompt**

# Use the intermediate representation and the schema links to generate the SQL queries for each of the questions.

Q: "Find the title of courses that have two prerequisites?"
Schema_links: [course.title,course.course_id = prereq.course_id]
A: Let's think step by step. "Find the title of courses that have two prerequisites?" can be solved by knowing the answer to the following sub-question "What are the titles for courses with two prerequisites?". The SQL query for the sub-question "What are the titles for courses with two prerequisites?" is SELECT T1.title FROM course AS T1 JOIN prereq AS T2 ON T1.course_id = T2.course_id GROUP BY T2.course_id HAVING count(*) = 2
So, the answer to the question "Find the title of courses that have two prerequisites?" is =
Intermediate_representation: select course.title from course where count ( prereq.* ) = 2 group by prereq.course_id SQL: SELECT T1.title FROM DB_Name.course AS T1 JOIN DB_Name.prereq AS T2 ON T1.course_id = T2.course_id GROUP BY T2.course_id HAVING count(*) = 2

...

Q: List all {Target_Features} information that satisfy following [Cohort Selection].
[Cohort Selection]: {Cohort_Selection}
Make PostgreSQL follow the order of the provided information, categories, type.
Schema_links: {Schema_links}
A: Let's think step by step.

**Self-correction prompt**

#### For the given question, use the provided tables, columns, foreign keys to fix the SQL. If correct, return as is.

Question: List all {Target_Features} information that satisfy following [Cohort Selection].
[Cohort Selection]: {Cohort_Selection}
Make PostgreSQL follow the order of the provided information, categories, type.
Schema_links: {Schema_links}
SQL Query: {sql_query}

#### Fixed SQL Query:
SELECT

## F.4   REACT

REACT (Yao et al., 2023) proposes a structured reasoning framework in which an agent takes appropriate actions based on observations from given environment to solve tasks. In our setting, the task involves generating proper SQL to get the user-requested dataset from a fixed database as shown in Listing 2. We extend this structure to both the Cohort and Feature Selection and Code Mapping tasks by formulating SQL generation as a sequence of reasoning steps. At each step, the model performs an action, and observes results from the database by predefined tool, enabling it to iteratively refine its reasoning toward solving the task. We use same prompt in Section F.1.

```python
from langchain_core.tools import tool
from langgraph.prebuilt import create_react_agent

def execute_query(query: str):
    """Use this to execute a query against the database."""

    try:
        db_observation = db_connector.connect(query)
```

```
    except Exception as e:
        return f"Error executing query: {str(e)}"

    if len(db_observation) > args.max_obsoutput_len:
        db_observation = db_observation[:args.max_obsoutput_len]

    return f"SQL Successfully executed. The example of {args.
        max_obsoutput_len} rows are as follows:\n{db_observation}"

def react_generation(prompt, llm_model):
    tools = [execute_query]
    react_agent = create_react_agent(model=llm_model, tools=tools)

    agent_inputs = {"messages": [("user", prompt)]}
    # print(agent_inputs)

    stream = react_agent.stream(agent_inputs, stream_mode="values",
        config={"recursion_limit": 20})

    response_list = print_stream(stream)
    api_run_count = str(response_list).count('AIMessage')
    observation_count = str(response_list).count('ToolMessage')
    final_result = response_list[-1]["messages"][-1].text()

    return final_result, api_run_count, observation_count
```

**Listing 2:** REACT interacting with database in Python

## G APPENDIX G. EMR-AGENT PROMPTS

### G.1 PROMPTS OF CFSA (COHORT AND FEATURE SELECTION AGENT)

The following provides the detailed prompts used for CFSA, as described in Section 3.2.

#### G.1.1 SCHEMA LINKING AND GUIDELINE GENERATION (MAPPING SCHEMA)

---

**Schema Linking and Guideline Generation (for Mapping Schema)**

Using [Database schema information], select all schema that are necessary to extract [Features].
- Please select exact table name(s) and column name(s) in [Database schema information].
- Follow the exact "Format" under [Notes] without any extra symbols, code delimiters.

[Notes]:
- Identify only definition schema with mapping information that can be used to extract patients of [Cohort Selection] with [Features].
- Exclude all tables that have actual numeric measurement (vital sign or lab test) columns.
- If identified tables have measurement unit information (not results), get all columns without result information.
- After listing the schema for each feature, provide a [Schema Guideline] in a paragraph of no more than 10 sentences, explaining the details of the columns (such as type or how to interpret the values).
- Output Format:

```
    Mapping Table: dbname.Table_A , Columns: Column_a, Column_b
    Mapping Table: dbname.Table_B , Columns: Column_1, Column_2
    [Schema Guideline]: (a paragraph of no more than 10 sentences)
```

[Schema Information]: {Schema_information}
[Evaluation Memo]: {Evaluation_Memo}
[Database Manual]: {Database_Manual}

---

[Cohort Selection]: {Cohort_Selection}
[Features]: {Feature_Selection}

### G.1.2 SCHEMA LINKING AND GUIDELINE GENERATION (FEATURE SCHEMA)

---

**Schema Linking and Guideline Generation (for Feature Schema)**

Using [Database schema information], select all schema that are necessary to extract [requested feature].
- Please select exact table name(s) and column name(s) in [Database schema information].
- Follow the exact "Format" under [Notes] without any extra symbols, code delimiters.

[Notes]:
- Get all schema (tables, columns) related to each element in [Features] and [Cohort Selection].
- After listing the schema for each element in [Features] and [Cohort Selection], provide a [Schema Guideline] in a paragraph of no more than 15 sentences,
explaining the details of the selected schema's columns (such as type or example values) and how to generate SQL to obtain patients from [Cohort Selection] with each [Features] and what it is missing to get the correct result.
- If necessary, utilize [Foreign Key] and [Mapping Table] from [Database schema information] when generating [Schema Guideline] for [Cohort Selection].
- Get patient's related year, date, time information such as admission date, birth date, etc.
- [Feature name] must be exactly same with [Feature].
- Output Format:

```
        [Feature name]
        Table Name: dbname.Table_A , Columns: Column_a, Column_b
        Table Name: dbname.Table_B , Columns: Column_1, Column_2
        [Feature name]
        Table Name: dbname.Table_A , Columns: Column_a, Column_b
        Table Name: dbname.Table_B , Columns: Column_1, Column_2
        ...
        [Schema Guideline]: (a paragraph of no more than 15 sentences)
```

[Schema Information]: {Schema_information}
[Evaluation Memo]: {Evaluation_Memo}
[Database Manual]: {Database_Manual}
[Cohort Selection]: {Cohort_Selection}
[Features]: {Feature_Selection}

---

### G.1.3 SQL SUFFICIENCY ASSESSMENT

---

**SQL Sufficiency Assessment**

You are an assistant tasked with evaluating the provided schema and guideline to determine if they are sufficient to support data extraction requirements.

Carefully review the following components:
- Original Schema: The schema before Schema Linking.
- Selected Schema: The schema and its guideline to assist to extract patients according to [Cohort Selection] with specified features [Target Features].
- Target Features: Specific features required for each patient. Note that names in [Target Features] are not always same in [Schema], do not assume value in schema.
- Cohort Selection: specifications for the configuration of patients to extract.
- Mapping Table: A table(s) and column(s) that contain mapping information of certain features, indicating details/definitions of certain features.

---

- Foreign Key: A foreign keys of the original schema.
- Error Feedback: If available, feedback from previously generated SQL queries indicating errors.
- Previous Observation (if provided): Previously observed information through SQL queries. Do not generate any SQL query that is already in [Previous Observation].

Task:
Assess whether the current [Selected Schema] and associated [Schema Guideline] are ENOUGH to extract the [Patients] according to [Cohort Selection] with [Target Features]. Classify your evaluation clearly into one of the following:
<need more information>: The [Selected Schema] and [Schema Guideline] are insufficient or require clarification.
        - Do not simply assume the names in [Target Features] and [Cohort Selection] are in [Selected Schema] and [Schema Guideline]. If you are not sure about the values, you need to first check or ask for the actual values that exist in the column (e.g., via 'SELECT DISTINCT column FROM table') before using them.
        - Only use a specific value in WHERE clauses if it is explicitly observed in the schema or query result, otherwise keep you position as <need more information>.
        - If [Error Feedback] exists and indicates issues, generate additional SQL queries to retrieve missing details.
<correct>: The provided [Selected Schema] and [Schema Guideline] are sufficient.

If you classified the schema as <need more information>,
- Generate SQL queries to retrieve the necessary additional details.
- If multiple queries are needed, separate each with ||.
- Do not generate SQL queries that retrieve entire tables — focus only on concise, targeted retrievals.
- If you need to use [Mapping Table] and [Foreign Key], please use them in the SQL queries.

Output Format: Provide your response exactly as below, without additional commentary or text:

<think>
[Clearly and concisely explain your reasoning behind the classifications based on the given information.]
</think>

<output>
<need more information> or <correct>
</output>

<SQL queries>
[If you classified the schema as <need more information>, based on you think process, [Schema Guideline] and [Additional Information], provide SQL queries to retrieve additional details from the schema using [Original Schema], [Mapping Table] and [Foreign Key]. If multiple queries are needed, separate each with ||. Note that the number of SQL queries should not exceed [Max SQL Search At Once]. Do not include any SQL query that is already in [Previous Observation].]
</SQL queries>

[Original Schema]:{Original_Schema}
[Selected Schema]:{Selected_Schema}
[Target Features]:{Target_Features}
[Cohort Selection]:{Cohort_Selection}
[Mapping Table]:{Mapping_Table}
[Foreign Key]:{Foreign_Key}
[Previous Observation]:{Previous_Observation}
[Error Feedback]:{Error_Feedback}

### G.1.4 DATA SUFFICIENCY CHECK

---

**Data Sufficiency Check**

You are an assistant to observe [SQL Observation] and find extra information to add to [Schema Guideline] to assist when generating SQL query for [Cohort Selection] patients with each of [Target Features].

Carefully review the following components:
- Original Schema: Includes tables, columns, and associated values before Schema Linking.
- Selected Schema: The schema and its guidelines chosen to extract patients according to [Cohort Selection] with specified features [Target Features].
- Target Features: Specific features required to extract for each patient.
- Cohort Selection: specifications for the configuration of patients to extract.
- SQL Observation: Results from executed SQL queries, provided as a dictionary (query-output pairs), offering further insights into [Original Schema] and possibly suggest more information to add to [Selected Schema]. The output could be an error message if the SQL query is failed. Keep in mind that the length of [SQL Observation] is limited to 20.
- Pre-Observation: Previously observed information.

Task:
- Select one of the below two options:
<Add info>: If you found something valuable information from [SQL Observation]
<No info>: If you found nothing valuable information from [SQL Observation]
- If you selected <Add info>, provide the gained information from [SQL Observation] in less than 5 sentences between <Add info> and </Add info>. The gained information should improve the [Schema Guideline] to extract the [Patients] according to [Cohort Selection] with [Target Features].

Output Format: Provide your response exactly as below, without additional commentary or text:

<think>
[Clearly and concisely explain your reasoning behind the classifications based on the given information in less than 5 sentences.]
</think>

<output>
<Add info> or <No info>
</output>

<Add info>
[Do not include information that is already in [Selected Schema] and [Schema Guideline]. Provide the gained information from [SQL Observation] in less than 6 sentences. This should be helpful to improve the [Schema Guideline] to extract the [Patients] according to [Cohort Selection] with [Target Features].]
</Add info>

[Original Schema]:{Original_Schema}
[Selected Schema]:{Selected_Schema}
[Target Features]:{Target_Features}
[Cohort Selection]:{Cohort_Selection}
[Previous Observation]:{Previous_Observation}
[SQL Observation]:{SQL_Observation}

---

### G.1.5 UPDATE SCHEMA LINKING AND SCHEMA GUIDELINE

---

**Update Schema Linking and Schema Guideline**

You are an assistant tasked with editing the [Schema Guideline] and [Schema] based on newly obtained [Additional Information]. Carefully review the following components:

- [Schema]: The original schema for [Target Features] and [Cohort Selection].
- [Schema Guideline]: The original schema guideline for [Target Features] and [Cohort Selection].
- [Additional Information]: New information gained from SQL Observation(s).
- [Target Features]: Specific features required to extract for each patient.
- [Cohort Selection]: specifications for the configuration of patients to extract.

Task:
- Make sure to update both [Schema Guideline] and [Schema] based on [Additional Information].
- Update the [Schema Guideline] and [Schema] based on [Additional Information] to support SQL query generation for extracting [Target Features] from the [Cohort Selection] patients.
- If [Additional Information] resolves previously unknown parts in [Schema Guideline], update them accordingly in [Schema Guideline].
- Provide the updated [Schema Guideline] no more than 15 sentences between <edited schema guideline> and </edited schema guideline>.
- Provide the updated [Schema] between <edited schema> and </edited schema> with the same format as the original [Schema] but with updated information such as column name, column type, column value (you can even add value examples), etc.
- If there is no need to update, provide the original [Schema Guideline] and [Schema].

Output Format: Provide your response exactly as below, without additional commentary or text:

<think>
[Explain your thought process clearly and concisely in no more than 5 sentences.]
</think>

<edited schema guideline>
[Edited Schema Guideline no more than 15 sentences]
</edited schema guideline>

<edited schema>
[Edited Schema]
</edited schema>

[Selected Schema]:{Selected_Schema}
[Schema Guideline]:{Schema_Guideline}
[Additional Information]:{Additional_Information}
[Target Features]:{Target_Features}
[Cohort Selection]:{Cohort_Selection}

---

### G.1.6 SQL GENERATION

---

**SQL Generation**

Q: Using the provided [Schema] with tables and columns and [Schema Guideline], write a PostgreSQL query to extract patients according to [Cohort Selection] with specified features [Target Features]. Output is only the SQL query as plain text. Do not include code delimiters

Follow these steps:
1. Select appropriate foreign keys(columns) provided in [Relation Information] to connect identified tables.

---

2. If necessary, use selected foreign key to make "JOIN". Do not use any other columns.
3. Ensure that each column referenced in the SELECT clause is present in the table alias used.
4. Use [Requested Features] to follow the sequence and format of '()' in [Requested Features] to generate the SQL query.
5. If some values are not visually understandable due to mapping code, add 'CASE' and 'WHEN' to replace the values with understandable values.
6. When writing WHERE conditions involving categorical values (e.g., gender, status), Do not assume specific values.
7. Only use a specific value in WHERE clauses if it is explicitly observed in the schema or query result.
8. When applying multiple inclusion/exclusion criteria, ensure that logically dependent conditions are ordered correctly.
- Do not reorder or drop dependent conditions; maintain logical dependencies when translating natural language criteria into SQL.
9. For all float values in the SQL output, cast them to ::float in the SELECT clause. If rounding is applied, first cast to numeric for ROUND(..., n) to work, then cast the result back to ::float if a float output is desired.
10. In SQL WHERE clauses, string comparison is case-sensitive. Use LOWER(), UPPER(), or adjust collation if you need case-insensitive matching.

SQL generate rule:
- Ensure that the SQL query only applies numeric comparisons (such as BETWEEN) on values that are safely converted to integers, thereby preventing type conversion errors.
- Always extract the Patient ID as-is (without deduplication, filtering, or counting) for the first column, exactly as it appears in the database.

Output Format: Provide your response exactly as below, without additional commentary or text:

<think>
[Clearly and concisely explain your reasoning behind the sql generation based on the given information.]
</think>

<SQL query>
[Write a PostgreSQL query to extract requested features of patients according to [Cohort Selection] with [Requested Features]]
</SQL query>

Feedback Note: [Previous Failed SQL] and [Error Feedback] are failed SQL and error feedback. Carefully exmaine [Error Feedback] and avoid [Previous Failed SQL] to generate correct SQL.
[Cohort Selection]:{Cohort_Selection}
[Target Features]:{Target_Features}
[Selected Schema]:{Selected_Schema}
[Schema Guideline]:{Schema_Guideline}
[Previous Failed SQL]:{Previous_Failed_SQL}
[Error Feedback]:{Error_Feedback}

### G.1.7 ERROR FEEDBACK

**Error Feedback**

You are an assistant that classifies SQL execution errors.

Given:
- Failed SQL: The query that failed.
- Selected Schema: Schema used to generate the query.
- Target: Intended data to extract.

- Error Feedback: Database error message.

Task: Analyze the provided information and classify the error as one of the following:
<syntax error>: SQL syntax is incorrect (e.g., missing keywords, misplaced clauses, invalid syntax).
<wrong schema>: Schema-related issue (e.g., referencing non-existent tables or columns, incorrect schema usage).

Output Format: Provide your response exactly as below, without additional commentary or text:

<think>
[Explain your thought process clearly and concisely in less than 6 sentences, highlighting why you chose this classification and exactly what factors caused the error.]
</think>

<error class>
<syntax error> or <wrong schema>
</error class>

[Selected Schema]:{Selected_Schema}
[Schema Guideline]:{Schema_Guideline}
[Cohort Selection]:{Cohort_Selection}
[Target Features]:{Target_Features}
[Failed SQL]:{Failed_SQL}
[Error Feedback]:{Error_Feedback}

## G.2  PROMPTS OF CMA (CODE MAPPING AGENT)

The following provides the detailed prompts used for CMA, as described in Section 3.3.

### G.2.1  SCHEMA LINKING AND GUIDELINE GENERATION (MAPPING SCHEMA)

**Schema Linking and Guideline Generation (for Mapping Schema)**

Using [Database schema information], select all schema that are necessary to extract [Features].
- Please select exact table name(s) and column name(s) in [Database schema information].
- Follow the exact "Format" under [Notes] without any extra symbols, code delimiters.

[Notes]:
- Identify only definition schema (table(s), column(s), and 3 sample values for each column) related to [Feature].
- Exclude columns that have actual measurement values (vital sign or lab test).
- The columns of definition sceham must include [Feature]'s information such as code, item number, name, abbreviation, etc.
- If identified definition table(s) have measurement unit information (not measurement value), get all the columns without actual measurement value information.
- After listing the schema for each feature, provide a [Schema Guideline] in a paragraph of no more than 10 sentences, explaining the details of the columns (such as type or how to interpret the values).
Output Format:

```
        Mapping Table: dbname.Table_A , Column: Column_a,
        Values: [value_1, value_2, value_3], Column: Column_b,
        Values: [value_1, value_2, value_3], Column: Column_c
        Mapping Table: dbname.Table_B , Column: Column_a,
        Values: [value_1, value_2, value_3], Column: Column_b,
        Values: [value_1, value_2, value_3], Column: Column_c
```

```
            [Schema Guideline]: (paragraph of no more than 10 sentences)

[Schema Information]: {Schema_information}
[Evaluation Memo]: {Evaluation_Memo}
[Database Manual]: {Database_Manual}
[Feature]: {Feature_Selection}
```

## G.2.2 SCHEMA LINKING AND GUIDELINE GENERATION (FEATURE SCHEMA)

---

**Schema Linking and Guideline Generation (for Feature Schema)**

Using [Database schema information], select all schema that are necessary to extract [requested feature].
- Please select exact table name(s) and column name(s) in [Database schema information].
- Follow the exact "Format" under [Notes] without any extra symbols, code delimiters.

[Notes]:
- Select all schema (tables, columns, and 10 sample values for each column) related to extract [Feature], including definition table(s) and measurement table(s) of [Feature].
- The selected schema must include tables such as definition table(s) and measurement table(s) of [Feature].
- Provide a [Schema Guideline] in a paragraph of no more than 5 sentences, explaining the details of the columns (such as type or how to interpret the values).
Output Format:
<selected schema>

```
Table Name: dbname.Table_A , Column: Column_a,
Values: [value_1, value_2, value_3, value_4, ..., value_10]
Table Name: dbname.Table_B , Column: Column_1,
Values: [value_1, value_2, value_3, value_4, value_5, ..., value_10]
```

</selected schema>

<schema guideline>
[Schema Guideline in a paragraph of no more than 5 sentences]
</schema guideline>

[Schema Information]: {Schema_information}
[Evaluation Memo]: {Evaluation_Memo}
[Database Manual]: {Database_Manual}
[Feature]: {Feature_Selection}

---

## G.2.3 FEATURE LOCATING

---

**Feature Locating**

Classify whether the [Feature] name is literally present in any column name(s) of [Selected Schema]. If necessary, use [Schema Guideline] to help you classify whether the [Feature] name is literally present in any column name(s) of [Selected Schema].
If the [Feature] name is literally present in any column name(s) (e.g., [Feature]: 'chris', and [Selected Schema] has column names 'Destin', 'tom', 'CHRIS'), return it as SchemaName.TableName.ColumnName between <featurecolumn> and </featurecolumn>.
- Matching should be case-insensitive, space-insensitive, and symbol-insensitive. Reasonable abbreviations are also accepted.
- Do not match semantic or contextual similarity. Only match if [Feature] name is a literal substring of the column name after removing case, space, and symbol differences.

---

- If more than one column name is present in [Selected Schema], return all of them in <feature column> separated by || as SchemaName.TableName.ColumnName || SchemaName.TableName.ColumnName || ...
- Never match based on content or examples of values in the column.
If the [Feature] name is not literally present in any column name(s) (e.g., [Feature]: 'chris', and [Selected Schema] has column names 'name', 'tom', 'Andy'), output <feature column>None</feature column>.
- If the [Feature] name only matches semantically or through contextual similarity, but not literally, output <feature column>None</feature column>.

Output Format: Provide exactly:
<think>
[Explain your thought process clearly and concisely in no more than 5 sentences.]
</think>

<feature column>
[SchemaName.TableName.ColumnName if [Feature] name is literally present in any column name(s) from [Selected Schema], or None if not.]
</feature column>

### G.2.4 CANDIDATE LISTING

**Candidate Listing**

Q: Using the provided Schema (tables, columns, values) and [Schema Guideline], generate a single PostgreSQL query to obtain columns 'unique feature identifier code (if exists)', 'feature name' and 'unit' from [Definition table].

To make a SQL query, follow these steps:
1. Identify tables that appear both in the [Feature Schema] and [Definition Schema]. Avoid using tables that are not in 'both' [Definition Schema] and [Feature Schema].
2. For identified table, ensure to obtain columns in the order of 'unique feature identifier code (if exists)','feature name' and 'unit'.
- Obtain 'unique feature identifier code' that represents feature types or items, but not row-level event-level IDs.
- Do NOT include the actual measurement value column.
3. If the table does not have a column about unit, look up the [Relation information] and [Feature Schema] to find any table that could provide the unit information via a foreign key relationship (e.g.,measurement id, machine id, etc.). Then JOIN that table to retrieve the correct unit column.
4. Use consistent aliasing for each table (e.g.,table AS alias) and ensure all aliases used in the SELECT clause are defined in the FROM clause.
5. Only use JOIN when necessary.
- Do not JOIN between each tables in [Definition Schema]. - Use JOIN only when there is a connection (foreign key) in [Relation Information].
6. Ensure the 'feature name' represents name of vital sign or lab test but not type or code number.
7. The order of the columns in the SELECT clause must be 'feature code number', 'feature name', and 'unit'.

Note for SQL formation:
1. Your final answer for each query must start from 'SELECT' (do not include any code fences or explanation).
2. Use DISTINCT to eliminate duplicate feature names. Return only one row per unique feature name.

3. When [Failed SQL] exists, carefully review the [Failed SQL] and [Error Feedback] to identify the cause of the failure and avoid the same mistake in the next SQL generation.

Output Format:
<think>
[Clearly and concisely explain your reasoning behind your SQL query generation.]
</think>

<SQL queries>
[SQL QUERY HERE]
</SQL queries>

### G.2.5 TARGET AND CANDIDATES MATCHING STRATEGY

Since there can be a large number of candidates, the LLM internally filters out those with low similarity to the target feature during the initial **Target and Candidates Matching** step. In this way, only the most relevant candidates are presented, rather than displaying all candidates and their probabilities. As described in Section 3.3, a user-defined threshold is applied in the second **Target and Candidates Matching** step to further filter candidates.

---

**First Target and Candidates Matching**

Compare each of the [Targeting Features] with each tuple from [Candidate Features] using your medical knowledge.
Assign similarity probabilities within a range of 0 to 100 for each pair, ensuring the comparisons reflect the degree to which each Candidate Feature aligns with the specific Targeting Feature.
Only include [Candidate Features] tuple(s) with similarity probabilities that is equal or higher than the specified Similarity Threshold.

Formatting Requirements:
1. Targeting Features: Each result must begin with the name of the Targeting Feature, followed by a colon (:).
2. Candidate Features and Probabilities: After the colon, include a dictionary where:
- Each Candidate Feature is a key (tuple format).
- The assigned probability (0 to 100 integer only) reflects how strongly the Candidate Feature belongs to the same category or type as the Targeting Feature.
- Only unique candidate feature tuples should be included (i.e., do not repeat the same candidate feature multiple times).
3. Key-Value Separators: Use double-pipes ‖ to separate key-value pairs inside the dictionary.
4. Separator: Use a semicolon (;) to separate results for each Targeting Feature.
5. No Additional Text: The output must strictly adhere to this format, and do not include code delimiters.

Example Input:
[Targeting Features]: Heart Rate
[Candidate Features]: [('C-reactive protein',), ('Pulse',), ('Serum Glucose',), ('SBP',)]
[Threshold]: 10
[Similarity Probabilities]:
('C-reactive protein',): 10‖ ('Pulse',): 90

[Targeting Feature]:{Targeting_Feature}
[Candidate Features]:{Candidate_Features}
[Threshold]:{User_defined_threshold}
[Similarity Probabilities]:

**Second Target and Candidates Matching**

Compare the [Targeting Feature] with each tuple in [Synonyms] using medical knowledge. Assign probabilities within a range of 0 to 100 for each pair, ensuring the comparisons reflect how strongly each Synonym belongs to the same category or type as the specific Targeting Feature.
Only include similarity probabilities that is equal or higher than the specified threshold.

Formatting Requirements:
1. Targeting Features: Each result must begin with the name of the Targeting Feature, followed by a colon (:).
2. Synonyms and Probabilities: After the colon, include a dictionary where:
- Each Synonym is a key (tuple format).
- The assigned probability (0 to 100) reflects how strongly the Synonym belongs to the same category or type as the Targeting Feature.
- Only unique synonym tuples should be included (i.e., do not repeat the same synonym multiple times).
3. Key-Value Separators: Use double-pipes (‖) to separate key-value pairs inside the dictionary.
4. No Additional Text: The output must strictly adhere to this format, and do not include code delimiters.

Example 1:
[Targeting Features]: CRP
[Synonyms]: ('C-reactive protein',), ('Pulse',), ('Serum Glucose',), ('SBP',) Similarity Threshold: 1
[Similarity Probabilities]: CRP: ('C-reactive protein',): 99‖ ('SBP',): 3

Example 2:
[Targeting Features]: Heart Rate
[Synonyms]: ('C-reactive protein',), ('Pulse',), ('Serum Glucose',), ('SBP',)
Similarity Threshold: 80
[Similarity Probabilities]: Heart Rate: ('Pulse',): 95

Example 3:
[Targeting Features]: SBP (mmHg)
[Synonyms]: ('Systolic Blood Pressure', 'mmHg'), ('Diastolic Blood Pressure', 'mmHg'), ('Heart Rate', 'bpm'), ('Serum Glucose', 'mg/dL') Similarity Threshold: 50
[Similarity Probabilities]: SBP: ('Systolic Blood Pressure', 'mmHg'): 98 ‖ ('Diastolic Blood Pressure', 'mmHg'): 60

[Target Feature]:{Target_Feature}
[Candidate Features]:{Candidate_Features}
[Threshold]:{User_defined_threshold}
[Similarity Probabilities]:

**Integration prompt**

Make PostgreSQL query to get {user_requested_event_stream_dataset} using [CFSA Generated SQL], [CMA Schema Linking], [CMA Schema Guideline] and [Selected mapping codes]. Make sure to satisfy [Note] to make optimized query for Large dataset.
[Note]
- Do not chage column name or alis, just use same SELECT information.
- Avoid Redundant Joins
- Use CTEs for Clarity & Indexing
- Push Filters Earlier

Also integrate two schema guidelines [CFSA Schema Guideline] and [Selected Mapping Code Guideline] in order to integrate all information and generate final correct PostgreSQL.

Use early reduction of data volume to optimize SQL query short. Before using JOIN, apply WHERE limit condition to get data faster. As possible, Place filter conditions at the top of the subquery.
Output is only one SQL query as plain text according to the output format.

Only select values that statisfy [Time condition] that means interval time between 'feature observation' and 'ICU admission' time. Do not select values that don't have time information.

[CMA Schema Linking] {cma_schema_linking}
[CMA Schema Guideline] {cma_schema_guideline}
[Selected mapping codes] {selected_mapping_codes}
[Target time range] {target_time_range}
[CFSA Generated SQL] {cfsa_generated_sql}
[CFSA Schema Guideline] {cfsa_schema_guideline}
[Selected Mapping Code Guideline]
Timeseries result about {Target_Feature} is '{selected_mapping_codes}' in
database, get feature result information that only about '{selected_mapping_codes}'.

Present your final output in the output format:
<think>
[Clearly and concisely explain your reasoning behind the sql generation based on the given information.]
</think>

<sql_query>
[The final generated PostgreSQL query to extract final_output_columns]
</sql_query>

Do not add any explanations or additional text outside of the specified output format.

# H APPENDIX H. USE OF LARGE LANGUAGE MODELS (LLMS)

To aid with writing and editing, we made limited use of LLM-based assistants (*e.g.*Claude). Their role was restricted to:

- Polishing grammar, style, and readability of paragraphs drafted by the authors.
- Summarizing longer drafts into shorter, more concise text upon author request.

No LLMs were used for generating research ideas, designing experiments, or producing results. All technical contributions, methods, and analyses were conceived and implemented entirely by the authors.

