# OpenReview forum: "EMR-AGENT: Automating Cohort and Feature Extraction from EMR Databases"
_ICLR.cc/2026/Conference — Submitted to ICLR 2026_

### Official Review · Reviewer_PcDo · 2025-10-20

**Soundness:** 2
**Presentation:** 3
**Contribution:** 3
**Rating:** 4
**Confidence:** 3

**Summary:**

This paper introduces an LLM-based agentic framework for extracting analysis cohorts from EHR-based datasets, specifically those from an ICU setting (MIMIC, eICU, SICdb). As there are no established evaluation sets for this task yet, the authors define several use cases to evaluate their agent.

**Strengths:**

- This paper develops a thorough agentic framework that matches expert workflows, including external database documentation and iterative querying of the data to identify necessary data items.
- The authors propose a standardised evaluation protocol for assessment, which is applied across three commonly used datasets, one of which is a very recent dataset published after their underlying LLM.
- The authors perform detailed ablation studies of each component of their framework.

**Weaknesses:**

- The paper is generally well written but I was missing some details on the main experiments (see Questions below)
- The evaluation sets for Cohort Selection are limited to 7 relatively simple scenarios. The ground truth comparisons to Harutyunyan and Sheikhalishahi are similarly limited to simple selections from those benchmarks. As a result, these scenarios are not representative of the actual breadth and depth of real-world cohort selections, which often involves complex combinations of observations (e.g., sepsis or ventilation).
- The evaluation sets for Code Mapping are limited to common labs, which are again among the simplest use cases for data harmonisation between ICU datasets. More complicated concepts such as medications or ventilation parameters are important for many research projects but not covered by the evaluation.
- It is hard to assess how impressive the performance of the agent is. The accuracy of ~30% for code mapping does not seem impressive but it is unclear whether this is driven by a bad agent or by any inherent uncertainty in the mapping task. A comparison to human mappers would help put those results in context.

**Questions:**

- I am unclear how accuracy was calculated for the Feature Selection task. For the example of Evaluation Set 1, is this just the proportion of cells in stay ID, gender, age, and length of stay that contained the same value as in the ground truth extract? Is this just for the subset of correctly identified patients? What if additional features/columns are returned, are those penalised or just ignored? What about permutations of the order of features?
- I am unclear how Code Mapping was evaluated. Was this just a multi-class prediction that assigned each possible feature (e.g., each entry in d_labitems and d_items in MIMIC) to either one of the 56 features and a bucket "other"? How can it be that "Ours w/o Candidate Matching "got a result of 0 for both F1 and bAcc?

**Details Of Ethics Concerns:**

It might be worth to clarify in the paper why sending data to Anthropic via the API complies with PhysioNet Data Use Agreements, which would usually preclude sharing data with third parties. This can be as easy as referencing https://physionet.org/news/post/gpt-responsible-use

---

> ### Author Response · Authors · 2025-11-21
>
> We thank the reviewer for the detailed and constructive feedback. We address each concern below.
>
> ---
>
> **Weakness 1,2: Missing Experimental Details & Limited Cohort Evaluation**
>
> We added clarification in *Appendix E*. Our evaluation follows a reproducible design. Since no standard exists for LLM-based cohort extraction, we selected seven scenarios that (1) permit unambiguous ground-truth verification across all three ICU datasets, (2) avoid institution-dependent clinical definitions, and (3) follow the evaluation scope used in prior structured-data benchmarks (Harutyunyan et al., 2019; Sheikhalishahi et al., 2019).
>
> While complex clinical conditions such as sepsis or ventilation are important, there is no consensus standard across datasets (e.g., Sepsis-1/2/3, varying ventilation protocols). Evaluating such cases would confound model performance with definition ambiguity. The seven scenarios serve as well-defined building blocks that EMR-AGENT can compose into multi-step logical filters. We added this limitation and planned extensions in *Appendix E*.
>
> ---
>
> **Weakness 3: Limited Code Mapping Evaluation**
>
> Our contribution lies not in covering more clinical concepts, but in automating a process that prior ICLR works, such as YAIB (ICLR 2024), implemented through handcrafted pipelines. Existing systems (YAIB, OMOP-based workflows such as BlendedICU) still depend on manual dataset-specific mappings, making cross-hospital onboarding labor-intensive.
>
> We focus on common lab concepts as reliable ground truth exists for them; medications and procedures require broader terminologies (RxNorm, SNOMED-CT) and expert-validated references that are not available across the three ICU datasets. EMR-AGENT is modular and can extend to such domains once consistent ground truth becomes feasible. This limitation and future extension plans are now in *Appendix E*.
>
> ---
>
> **Weakness 4: Interpretation of the ~30% Code Mapping Accuracy**
>
> Ground truth was built through multi-expert consensus, ensuring accuracy reflects validated mappings rather than heuristics. However, as the reviewer notes, raw accuracy alone does not capture the practical value of the agent for real EMR onboarding.
>
> A key distinction in this task is the disparity between low-cost and high-cost human operations. Verifying a suggested mapping (TP/FP) is fast and inexpensive, while recovering a missing mapping or confirming true absence (FN/TN) requires clinicians to manually search thousands of raw codes, an operation that dominates real-world preprocessing time. Thus, the practical question is not the absolute accuracy, but **how much manual labor the agent reduces**.
>
> As quantified in **Appendix D: Table D.2**, EMR-AGENT reduces high-cost operations from **933 to 399 (57%↓)**, far exceeding PLUQ and ReAct (5-17%). This measure is robust because it relies only on the universally valid asymmetry that FN/TN/Remaining require extensive manual search, whereas TP/FP do not. To avoid trivial over-predicting, the metrics must be interpreted with predictive accuracy (Table 1), preventing degenerate solutions.
>
> Thus, the ~30% raw accuracy reflects the intrinsic difficulty of harmonizing highly heterogeneous, hospital-specific code systems, but the **57% reduction in high-cost expert labor** demonstrates meaningful practical benefit: EMR-AGENT automates a portion of a process that previously required fully manual inspection while keeping clinicians in full control of final validation.
>
> ---
>
> **Q1: Feature Selection Accuracy Calculation**
>
> Feature Selection evaluation proceeds in two stages:
> 1. **Cohort correctness:** F1 over matched ICU stays.
> 2. **Feature correctness:** accuracy of requested feature values only for correctly matched patients.
>
> Column order is ignored (not clinically meaningful), and additional columns are not penalized. However, missing a requested feature (e.g., absent or misnamed column) yields zero credit for that feature. Evaluation compares ground-truth values only for the user-requested features and only for the correctly matched patients. We clarified this in Section 4.3.
>
> ---
>
> **Q2: Code Mapping Evaluation Method**
>
> Code mapping is treated as binary classification for each concept.
> **TP:** mapping in ground truth
> **FP:** produced but not in ground truth
> **FN:** ground-truth mapping missed
> **TN:** concept correctly identified as absent
> F1 and balanced accuracy are computed from these. Without Candidate Matching, the agent cannot search large tables or identify candidate rows, leading to 0 F1. This validates the module’s necessity.
>
> ---
>
> **Ethics Concerns**
>
> PhysioNet’s official guidance (GPT Responsible Use, *Item 4*) explicitly lists *Anthropic* as an approved provider. As previously noted in **Section 5** and now reiterated in the **Ethics** section, our use of Claude-3.5 fully complies with the PhysioNet DUA.
>
> ---
>
> We thank the reviewer again for the constructive feedback. The added clarifications and appendices strengthen the work.

---

### Official Review · Reviewer_Upsx · 2025-10-26

**Soundness:** 3
**Presentation:** 3
**Contribution:** 3
**Rating:** 4
**Confidence:** 3

**Summary:**

The paper introduces EMR-AGENT, an agentic framework to replace manual rule writing with dynamic language-model driven interaction to extract and standardize structured clinical data. The authors construct a standardized evaluation protocol and codebase for EMR preprocessing. The approach is evaluated through component-level ablations, comparisons with other LLM-based approaches, and evaluation on a held-out EMR database.

**Strengths:**

The paper is clear and well-written, and claims are properly backed by evidence and/or results.

The work seems significant in its application area, and provides strong solutions for described problems and limitations of existing work.

Experiments are sound and transparent, components of the approach are properly ablated, results are averaged over repeated trials, and the appendix contains sufficient details on specific prompts, benchmark construction process, and relevant reproduction details for the baselines.

**Weaknesses:**

While the work seems significant in its application area, it would likely be better suited for an application-specific journal, conference, or relevant workshop.

**Main Reasoning**

- While the described approach may be novel for EMR preprocessing specifically, similar approaches featuring described components such as iterative refinement, error handling, dynamic contexts featuring manuals or metadata, etc. are common and in some cases more comprehensive and generalizable in other applications of agentic LLMs such as search, literature review, or software engineering.
- The described prompts and steps are specific to EMR preprocessing, which is a relatively narrow application of a specific field.
- There are no other contributions such as additional training or fine-tuning approaches that would make the work a better fit.

While these reasons on their own are not necessarily grounds for rejection, I feel that in aggregate the paper is not a strong fit for ICLR.

**Questions:**

Given that my concerns are not with the execution or quality of the work, but mainly its relevance to ICLR specifically, I am open for discussion with the authors on my concerns outlined in the weaknesses section above.

---

> ### Author Response · Authors · 2025-11-21
>
> Thank you for raising the venue-fit concern. We respectfully argue that our contribution aligns well with the growing body of ICLR research on clinical data infrastructure and structured-data reasoning. Recent ICLR works such as **YAIB (ICLR 2024)**[1] and **ACES (ICLR 2025)**[2] demonstrate that EMR data infrastructure and schema standardization are already active topics within the community. However, **all existing works still rely on handcrafted, dataset-specific preprocessing pipelines**. They assume that schema interpretation, cohort extraction, and code harmonization have already been manually completed by domain experts. **None of them address AI-based automated EMR preprocessing.**
>
> Our work is, to our knowledge, **the first to automate this missing upstream stage** in the clinical structured-data setting using an agentic LLM framework. We view this study as a timely extension to the line of ICLR work on EMR data infrastructure, addressing a preprocessing step that prior ICLR papers do not automate. Moreover, the ICLR 2025 Call for Papers[3] explicitly includes healthcare and biological applications, and recent ICLR venues have repeatedly accepted EMR-based data modeling and LLM-agent healthcare applications, indicating that this domain is well within scope.
>
> **For clarity:**
> - **YAIB (ICLR 2024)** provides tools for cohort selection and code mapping but relies on **handcrafted pipelines tied to specific ICU datasets**.
> - **ACES (ICLR 2025)** offers configuration-driven extraction but still requires **manual specification of dataset-specific predicates** and assumes **MEDS/ES-GPT-compliant data formats**.
>
> Several reviewers also recognized the relevance of this direction. Reviewer SGSC noted that our paper "introduces the first agent-based, LLM-driven framework for EMR preprocessing" and mentioned the "comprehensive experimental design and valuable benchmark." Reviewer Ufas emphasized that EMR-AGENT "reframes extraction as an interactive reasoning problem," marking a **conceptual advance** over rule-based frameworks such as YAIB and ACES. These assessments support that EMR data infrastructure is already ICLR-relevant, and that our work fills the missing upstream stage by introducing the **first agentic LLM system for EMR preprocessing directly from raw hospital databases.**
>
> [1] Robin van de Water et al., ICLR 2024.
> [2] Justin Xu et al., ICLR 2025.
> [3] https://iclr.cc/Conferences/2025/CallForPapers

---

> > ### Comment · Reviewer_Upsx · 2025-11-21
> >
> > Thank you for your detailed response an relevant citations, especially the opinion of the area chair on the ACES submission at ICLR 2025, responding to similar concerns, has convinced me to increase my rating.

---

### Official Review · Reviewer_Ufas · 2025-10-26

**Soundness:** 3
**Presentation:** 3
**Contribution:** 3
**Rating:** 6
**Confidence:** 3

**Summary:**

This paper presents EMR-AGENT, a large language model (LLM)-driven framework that automates preprocessing of Electronic Medical Records (EMRs)—specifically cohort selection, feature extraction, and code mapping—tasks that traditionally rely on labor-intensive, database-specific SQL pipelines. The proposed system replaces manual rule-writing with an agent-based architecture that dynamically interacts with EMR databases through SQL queries, observes query outputs, and reasons over schema and documentation to iteratively refine its extraction strategy. EMR-AGENT is composed of two agents: the Cohort and Feature Selection Agent (CFSA) and the Code Mapping Agent (CMA). Both agents perform schema linking and guideline generation before engaging in SQL-based exploration, error correction, and iterative refinement.

To evaluate the framework, the authors build a new benchmark suite called PreCISE-EMR, covering three major ICU datasets—MIMIC-III, eICU, and SICdb—with tasks measuring both seen (MIMIC-III, eICU) and unseen (SICdb) schemas. Results show that EMR-AGENT significantly outperforms existing Text-to-SQL baselines such as PLUQ, EHR-SeqSQL, and DIN-SQL across all databases. CFSA achieves up to 0.94 F1 on MIMIC-III and 0.81 on unseen SICdb, while CMA attains 0.65 F1 on eICU. Ablations confirm that interactive SQL observation and schema-guided reasoning are critical for performance, and external documents (manuals, memos) greatly enhance robustness. The system generalizes across multiple LLM backbones, with Claude-3.5-Sonnet and Claude-3.7-Sonnet performing best, and demonstrates the feasibility of replacing rigid, expert-driven EMR pipelines with flexible, language model-based automation

**Strengths:**

- EMR-AGENT reframes cohort and feature extraction as an interactive reasoning problem rather than static rule application, marking a clear conceptual advance over existing harmonization frameworks such as YAIB, ACES, and Clairvoyance, which rely on handcrafted mappings.
- The paper carefully decomposes the agent workflow into interpretable modules (Schema Linking, SQL-based Observation, Error Feedback, and Candidate Matching). The visual diagrams (Fig. 2a–b, p. 4) make clear how these stages correspond to classical data engineering steps, lending transparency and auditability to the LLM-driven process.
- The authors do not merely report end-task performance but construct a full benchmarking environment (PreCISE-EMR)—with PostgreSQL instances, human-curated ground-truth mappings, and evaluation code—establishing a foundation for future reproducible research.
- The model maintains high accuracy across MIMIC-III, eICU, and SICdb, despite the latter being unseen during pretraining. This is particularly impressive given that schema heterogeneity is one of the hardest obstacles in real-world EMR integration.
- The benchmark includes human clinical oversight (two physicians, two nurses, one informatics expert), strengthening the validity of the evaluation results.
- By adapting leading Text-to-SQL models (PLUQ, EHR-SeqSQL, DIN-SQL, REACT) to the same schema environment, the authors show that naive translation systems fail dramatically (F1 ≤ 0.1 on some datasets), reinforcing the claim that dynamic, context-aware exploration is essential.
- The evaluation in Table 4 confirms that performance scales with reasoning capacity: Claude-3.5-Sonnet > Claude-3.5-haiku > open-source Llama3.1 and Qwen2.5, highlighting the need for reasoning-rich models while offering an efficient option for constrained settings.

**Weaknesses:**

- While EMR-AGENT automates query generation and schema reasoning, it stops short of true end-to-end preprocessing. Manual approval is still implicitly required to verify final cohorts and mappings, and the paper does not quantify human-in-the-loop correction effort.
- Even though CMA outperforms baselines, absolute F1 values (≈ 0.5) remain modest, suggesting that vocabulary harmonization remains an open challenge, especially for free-text or ambiguous concept labels.
- Cite the work of MEDS which has done extensive work to create a data schema for EMR/EHR (https://dl.acm.org/doi/abs/10.1145/3711896.3737608?casa_token=MkA4rBEmTXMAAAAA:_fER13VZVnNzAtiZ6GE2dgZIvmIGmhsObsrNj1u9zx1oamEj4-YU-emnUZZVyOX3OVAzvMPefRkv, https://openreview.net/forum?id=IsHy2ebjIG)
- While the framework claims to replace manual rule writing, the paper provides no measurement of time saved or human equivalence benchmarks (e.g., accuracy vs. expert engineer).
- Iterative querying over large databases could be computationally expensive, especially under tight token limits. The paper does not estimate query cost or latency across agents.

**Questions:**

- How does EMR-AGENT ensure semantic fidelity between user intent and generated SQL queries in ambiguous natural language inputs?
- Can you quantify the average number of SQL iterations or corrections per task, and how this correlates with performance?
- What mechanisms prevent hallucinated schema entities or unsafe queries (e.g., full-table scans on large hospital databases)?
- How would EMR-AGENT handle an EMR system with missingness or limited column metadata? does performance degrade gracefully or fail catastrophically?

---

> ### Author Response · Authors · 2025-11-21
>
> **Weakness 1,4: Quantifying Human-in-the-Loop Effort**
>
> We agree that full clinical verification is always required; automation cannot replace clinicians in medical settings. Our goal is to reduce expert workload when onboarding a new EMR. Without automation, clinicians must manually interpret unfamiliar schemas and search thousands of raw codes to identify valid mappings.
>
> EMR-AGENT automates these steps: Schema Linking identifies relevant fields, CFSA generates cohort extraction SQL, and CMA narrows thousands of raw codes to a small set, automating a stage that prior ICLR works such as YAIB (ICLR 2024) and ACES (ICLR 2025) handled through handcrafted rules. We quantify the expert workload that remains after automation using **Appendix D**’s strict criteria, which map model errors to clinician intervention.
>
> In CFS, even a **single added or missing patient** in the cohort output, or **a single incorrect cell** in any extracted feature (age, LOS), counts as an error requiring review (see **Appendix D.1**). In CM, the high-cost portion of expert labor corresponds to **FN**, **TN**, and **Remaining** concepts, requiring clinicians to manually search large raw-code spaces. As highlighted in **Appendix D.2**, these operations dominate real-world preprocessing effort, while TP/FP verifications incur negligible cost. To avoid trivial over-predicting, labor metrics must be interpreted with predictive accuracy (Table 1), preventing degenerate solutions.
>
> To make the cost structure explicit, we added **Appendix D.1 (CFS Error Counts)** and **Appendix D.2 (CM High-Cost Operations)**. Across 210 CFS tasks, EMR-AGENT reduces logic errors from 336-416 to 155 (54-63% reduction). In CM (933 checks), high-cost operations drop from 933 to 399 (57%), far exceeding PLUQ/ReAct (5-17%). Importantly, this metric requires no assumptions about annotation speed or expert time; it leverages the intrinsic asymmetry that FN/TN/Remaining require costly manual search, whereas TP/FP require only simple verification.
>
> Overall, **Appendix D** shows EMR-AGENT greatly reduces expert labor.
>
> ---
>
> **Weakness 5: Computation Cost and Latency**
>
> We added **Appendix C**, which reports the average LLM calls and runtime across all methods for both CFS (**Table C.1**) and CM (**Table C.2**) tasks. EMR-AGENT consistently uses **fewer calls than ReAct** while achieving much higher accuracy, showing that performance improvements do not result from excessive querying. Compared with lightweight single-shot methods (PLUQ, SeqSQL, DinSQL), EMR-AGENT uses moderately more calls but yields substantially higher F1. The additional calls originate from SQL-Observation and Error-Feedback. Ablations in **Table 3** show that removing SQL-Observation or Error-Feedback severely degrades accuracy, indicating that these iterations provide functional benefits beyond naive repeated querying. Overall, EMR-AGENT offers a favorable trade-off: **agent-level accuracy with competitive cost**.
>
> ---
>
> **Weakness 2,3: Other Reviewer Comments**
>
> **On modest CMA F1:** CM is inherently difficult due to heterogeneous local vocabularies. The practical benefit lies in reducing high-cost errors: CMA lowers FN/TN/Remaining by **50-59%** (Table D.2), substantially reducing manual vocabulary harmonization workload, even with modest absolute F1.
> **MEDS citation:** Reference has been added.
>
> ---
>
> **Q1. Semantic fidelity under ambiguous language**
>
> In CM, semantic drift is controlled by anchoring each concept to OMOP. In CFSA, ambiguity is resolved through a cycle (Schema Guideline → SQL Observation → Error Feedback). Any mismatch detected during Error Feedback regenerates the guideline, ensuring that the final SQL faithfully reflects user intent (further clarified in **Section 3.2**).
>
> ---
>
> **Q2. SQL iterations and their effect on performance**
>
> Figure 3 shows SQL-Observation iterations increase when metadata is removed (e.g. 5.07→7.57 in MIMIC-III, 4.19→6.85 in eICU, 2.98→7.81 in SICdb), reflecting compensatory querying under schema uncertainty. **Table 2** shows removing SQL-Observation or Error-Feedback greatly reduces performance, confirming that deeper iteration improves accuracy.
>
> ---
>
> **Q3. Preventing schema hallucination and unsafe queries**
>
> Schema Linking restricts both agents to whitelisted tables/columns. Error Feedback eliminates hallucinated SQL: syntactic errors trigger regeneration, while semantic mismatches (missing columns, empty outputs, etc.) return the agent to guideline regeneration. CMA enforces DISTINCT; CFSA limits SQL-Observation values; and all queries use row limits to prevent unsafe operations.
>
> ---
>
> **Q4. Handling missing metadata**
>
> **Table 3** shows that removing metadata causes only moderate degradation; EMR-AGENT still outperforms baselines that use full metadata (Table 1). Figure 3 shows compensatory SQL-Observation increases, enabling graceful degradation rather than collapse.

---

> > ### Comment · Reviewer_Ufas · 2025-11-25
> > **Thanks you Authors**
> >
> > Thanks to the authors for taking the time to address my concerns. I have acknowledged that I have read the rebuttal and decide to keep my score.

---

### Official Review · Reviewer_SGSC · 2025-11-01

**Soundness:** 2
**Presentation:** 3
**Contribution:** 2
**Rating:** 4
**Confidence:** 3

**Summary:**

This paper introduces EMR-AGENT, agent-based framework for automating preprocessing of EMRs. The framework replaces manually crafted, schema-specific rules with LM-driven agents that iteratively query, observe, and reason over database schemas and documentation. Authors also introduce a new benchmark suite built on three EMR databases to assess performance under both seen and unseen schema settings. Ablation studies confirm the importance of schema-guided linking, SQL-based observation, and error feedback.

**Strengths:**

- Introduces the first agent-based, LLM-driven framework for EMR preprocessing
- Pretty comprehensive experimental design, including multi-database evaluation, detailed ablation studies, and model generalization tests.
- Empirical evidence from MIMIC-III, eICU, and SICdb
- Benchmark contribution is valuable.

**Weaknesses:**

- Limited interpretability analysis into how agents make reasoning decisions or handle ambiguous schema mappings
- No comparison of computational costs against other baselines unless i am missing something since agent-based methods are very expensive when recursive

**Questions:**

- How does EMR-AGENT handle ambiguous or missing schema documentation in entirely novel hospital EMRs beyond the three tested datasets?
- Could the framework be adapted for continuous schema evolution (e.g., schema drift in live hospital systems)?
- Have the authors evaluated whether EMR-AGENT’s extracted data lead to comparable or superior downstream model performance (e.g., mortality prediction) compared to manual preprocessing pipelines?

---

> ### Author Response · Authors · 2025-11-21
>
> **Weakness 1: Limited interpretability analysis**
>
> Thank you for the feedback. While deep interpretability is not the main focus of EMR-AGENT, our objective is to automate the EMR preprocessing stage that prior ICLR works such as YAIB (ICLR 2024) and ACES (ICLR 2025) implemented through handcrafted, non-scalable rules.
>
> To clarify how the agent makes decisions, we added **Appendix B**, which provides two step-by-step walkthroughs (one CFSA and one CMA) illustrating key steps such as schema linking, guideline generation, SQL-Observation, and Error Feedback, and how these components interact when resolving ambiguous or incomplete schema information. These examples show the full reasoning chain rather than isolated snippets.
>
> Quantitatively, **Table 2** shows that removing Schema Guideline, SQL-Observation, or Error-Feedback causes large performance drops, indicating that these iterative steps provide functional benefits beyond naive recursion.
>
> ---
>
> **Weakness 2: Computation cost and latency**
>
> We added **Appendix C**, which reports the average LLM calls and runtime across all methods for both CFS (**Table C.1**) and CM (**Table C.2**) tasks. EMR-AGENT consistently uses fewer calls than ReAct while achieving much higher accuracy, showing that performance improvements do not result from excessive querying.
> Compared with lighter single-shot or fixed-step baselines (PLUQ, SeqSQL, DinSQL), EMR-AGENT uses moderately more calls but achieves substantially higher scores across all EMRs. These additional calls mainly arise from SQL-Observation and Error-Feedback. Ablations in **Table 3** show that removing SQL-Observation or Error-Feedback severely degrades accuracy, indicating that these iterations provide functional benefits beyond naive repeated querying. Overall, EMR-AGENT offers a favorable trade-off: **agent-level accuracy with competitive cost**.
>
> ---
>
> **Q1: Missing schema documentation in novel hospitals**
>
> This scenario is already captured in our evaluation. **SICdb is a fully novel EMR**, created after all LLM pretraining cutoffs, functioning as a realistic “previously unseen hospital.” Despite having no exposure to SICdb’s schema, EMR-AGENT performs strongly across tasks (Tables 1 and 2), demonstrating its ability to generalize beyond memorized schemas.
>
> Additionally, removing schema documentation in **Table 3** shows only limited degradation, and EMR-AGENT without documentation still exceeds baselines that *do* have documentation (Table 1). This empirically supports the model’s robustness in environments where documentation is missing.
>
> ---
>
> **Q2: Adaptation to live hospital systems**
>
> Yes, EMR-AGENT is naturally compatible with schema drift. CFSA regenerates the Schema Guideline at every run using the current database snapshot and uses SQL-Observation to update schema understanding with real outputs. Newly added fields or changes in value formats are immediately reflected without manual adjustments.
>
> CMA similarly detects and re-evaluates new candidate fields through iterative matching, without requiring predefined mappings. These closed-loop mechanisms allow EMR-AGENT to maintain correctness and coverage under continuous schema evolution in live hospital systems.
>
> ---
>
> **Q3: Superiority of extracted data**
>
> Our goal is not to outperform handcrafted pipelines in downstream clinical prediction but to **eliminate the manual, institution-specific preprocessing burden** that prior ICLR works such as YAIB (ICLR 2024) and ACES (ICLR 2025) relied on. When the same feature definitions are used, downstream model accuracy depends primarily on the clinical variables themselves rather than the method used to extract them.
>
> The contribution of EMR-AGENT lies in providing consistent and reproducible preprocessing that remains robust to missing documentation and significantly reduces human effort across institutions. To make this explicit, we additionally included **Appendix D**, which quantifies the reduction in human-in-the-loop workload across CFS and CM. Evaluating relative downstream performance is therefore orthogonal to the core contribution of this work.

---

### Author Response · Authors · 2025-12-01

## **General Response**
Dear Area Chairs and reviewers,
We sincerely appreciate your effort and consideration in reviewing our manuscript. Below is a concise summary of our contribution, the strengths recognized by reviewers, and how all major concerns were addressed during rebuttal and discussion.

---

### **[1] Summary of the Paper**
EMR-AGENT is the **first AI-based, LLM-driven agent framework that automates EMR preprocessing** (cohort extraction, feature extraction, and code mapping) directly from raw, schema-divergent hospital databases. The framework consists of two agents: the Cohort & Feature Selection Agent (CFSA) and the Code Mapping Agent (CMA).
**While prior ICLR works (YAIB 2024, ACES 2025) still rely on handcrafted rules or require pre-standardized input formats, which still depend on substantial manual effort, EMR-AGENT performs fully agentic, interactive preprocessing** directly on raw schemas, iteratively generating the necessary SQL for all tasks.
Across the three EMR databases (MIMIC-III, eICU, and the unseen-schema SICdb), EMR-AGENT consistently outperforms existing baselines (Table 1). Ablation studies further confirm that each agentic component is functionally necessary for robust performance (Table 2).

---

### **[2] Strengths Highlighted by Reviewers**
- **Novelty & Significance.** The first agent-based LLM framework for EMR preprocessing, viewed by reviewers as a clear conceptual advance over rule-based EMR pipelines and well-aligned with ICLR’s existing EMR-infrastructure line (YAIB, ACES). `[SGSC, Ufas]`
- **Clear & Modular Design.** Reviewers emphasized the transparent pipeline (Schema Linking/Guideline, SQL Observation, Candidate Matching, Error Feedback). `[Ufas, PcDo, SGSC]`
- **Comprehensive Evaluation.** Multi-database assessment on both seen (MIMIC-III, eICU) and unseen (SICdb) schemas, with baselines comparison and detailed ablations. `[SGSC, Ufas, Upsx, PcDo]`
- **Benchmark Quality.** PreCISE-EMR viewed as valuable and well-constructed. `[SGSC, Ufas]`

---

### **[3] Weaknesses Raised by Reviewers and How They Were Addressed**
**(1) Venue Fit - `[Upsx]`**
We clarified that EMR data infrastructure is already an active ICLR topic (YAIB 2024, ACES 2025), and that EMR-AGENT automates the upstream preprocessing stage those works assume is handcrafted.
→ **Reviewer `Upsx` accepted the clarification and increased their rating (4 → 6).**

**(2) Computation Cost & Efficiency - `[SGSC, Ufas]`**
Appendix C now shows that EMR-AGENT delivers the highest accuracy while maintaining competitive cost: it uses fewer LLM calls than ReAct in CFS and more in CMA, but with substantially higher accuracy in both cases. Fixed-step baselines use fewer calls but perform far worse. Ablations confirm that the call-increasing components (SQL Observation, Error Feedback) are essential for correctness (Table 2).
→ **Reviewer `Ufas` acknowledged that these clarifications addressed their concerns.**

**(3) Human Effort Reduction - `[Ufas, PcDo]`**
Appendix D now shows substantial reductions in human workload: EMR-AGENT cuts high-cost Code Mapping (CM) operations by 57% (933 → 399) and reduces Cohort & Feature Selection(CFS) logic-error cases by 54-63% compared to baselines. These gains translate directly into fewer SQL-debugging interventions and far less manual code search for clinicians.
→ **Reviewer `Ufas` acknowledged that these clarifications addressed their concerns.**

**(4) Interpretability & Transparency - `[SGSC]`**
Appendix B now includes concise CFSA and CMA case studies that illustrate how the agent resolves ambiguous schema mappings through targeted inspection and iterative refinement, directly addressing the reviewer’s request for clearer reasoning transparency.

**(5) Limited Evaluation Scenarios - `[PcDo]`**
Appendix E clarifies that we use only scenarios with unambiguous cross-dataset ground truth. Complex conditions (e.g., sepsis, ventilation) lack consistent definitions across EMRs.

**(6) (Common Question) Robustness to Missing Metadata - `[SGSC, Ufas]`**
Table 3 shows that EMR-AGENT experiences only moderate degradation when metadata is removed, and still outperforms baselines that rely on full metadata (Table 1 and 3). This directly addresses both reviewers’ questions regarding robustness under missing or incomplete documentation.
→ **Reviewer `Ufas` acknowledged that these clarifications addressed their concerns.**

---

We hope this summary helps you quickly understand how reviewer concerns were comprehensively addressed. Thank you again for your time and consideration.

Warm regards,

Authors

---

### Meta-Review · Area_Chair_g39Z · 2026-01-09

**Summary:**

While the paper targets an important practical problem and the rebuttal addressed several implementation and clarity concerns,  there are some concerns that remain: (i) the core technical contribution is largely an application-specific integration of known agentic and Text-to-SQL techniques, with limited methodological novelty; (ii) the evaluation is intentionally restricted to simple, unambiguous scenarios and common lab concepts, limiting its representativeness of real-world EMR preprocessing (iii) the work is heavily domain-specific and system-oriented, making it an unclear fit for ICLR.

**Reviewer Concerns:**

Clarification of evaluation

**Reviewer Scores:**

unchanged

---

### Decision · Program_Chairs · 2026-01-26

Reject